# Immunization with V987H-stabilized Spike glycoprotein protects K18-hACE2 mice and golden Syrian hamsters upon SARS-CoV-2 infection

Carlos Ávila-Nieto[1,13], Júlia Vergara-Alert[2,3,13], Pep Amengual-Rigo[4,13], Erola Ainsua-Enrich[1], Marco Brustolin[2,3,12], María Luisa Rodríguez de la Concepción[1], Núria Pedreño-Lopez[1], Jordi Rodon[2,3], Victor Urrea[1], Edwards Pradenas[1], Silvia Marfil[1], Ester Ballana[1,5,6], Eva Riveira-Muñoz[1], Mònica Pérez[2,3], Núria Roca[2,3], Ferran Tarrés-Freixas[1,2,3], Guillermo Cantero[2], Anna Pons-Grífols[1], Carla Rovirosa[1], Carmen Aguilar-Gurrieri[1], Raquel Ortiz[1], Ana Barajas[1], Benjamin Trinité[1], Rosalba Lepore[4], Jordana Muñoz-Basagoiti[1], Daniel Perez-Zsolt[1], Nuria Izquierdo-Useros[1,5,6], Alfonso Valencia[4,7], Julià Blanco[1,5,6,7], Victor Guallar[4,8], Bonaventura Clotet[1,5,6,7,9,10], Joaquim Segalés[2,10,11] ✉ & Jorge Carrillo[1,5,6] ✉

Safe and effective severe acute respiratory syndrome coronavirus 2 (SARS-CoV-2) vaccines are crucial to fight against the coronavirus disease 2019 pandemic. Most vaccines are based on a mutated version of the Spike glycoprotein [K986P/V987P (S-2P)] with improved stability, yield and immunogenicity. However, S-2P is still produced at low levels. Here, we describe the V987H mutation that increases by two-fold the production of the recombinant Spike and the exposure of the receptor binding domain (RBD). S-V987H immunogenicity is similar to S-2P in mice and golden Syrian hamsters (GSH), and superior to a monomeric RBD. S-V987H immunization confer full protection against severe disease in K18-hACE2 mice and GSH upon SARS-CoV-2 challenge (D614G or B.1.351 variants). Furthermore, S-V987H immunized K18-hACE2 mice show a faster tissue viral clearance than RBD- or S-2P-vaccinated animals challenged with D614G, B.1.351 or Omicron BQ1.1 variants. Thus, S-V987H protein might be considered for future SARS-CoV-2 vaccines development.

Severe acute respiratory syndrome coronavirus 2 (SARS-CoV-2) is the etiological agent of coronavirus disease 2019 (COVID-19), an infectious disease that emerged at the beginning of December 2019[1,2] in Wuhan, China. COVID-19 was declared a pandemic by the World Health Organization (WHO) on March 11, 2020. As of November 30th 2023, over 770 million COVID-19-confirmed cases and 6.98 million deaths have been registered (WHO Coronavirus (COVID-19) Dashboard). Since the beginning of the pandemic, there has been a global effort to expedite the development of vaccines that could help control the COVID-19 pandemic. While more than 180 vaccine projects started

being developed in 2020[3], four vaccines (Pfizer/BioNTech, Moderna, AstraZeneca and Janssen) received emergency use authorization by the Food & Drug Administration (FDA) and/or the European Medicines Agency (EMA) in record time. These four vaccines are based on the SARS-CoV-2 Spike (S) glycoprotein, since it is the main target of neutralizing antibodies (NAbs)[4,5]. In fact, previous studies showed that immunizations with SARS-CoV and Middle East respiratory syndrome coronavirus (MERS-CoV) whole S glycoproteins or their subunits (e.g., receptor binding domain (RBD)) protected from disease development in animal models[6,7].

The SARS-CoV-2 S glycoprotein is a homotrimeric protein and each monomer is composed of two subunits: the S1 surface/external subunit and the S2 membrane anchor domain. While S1 binds to the angiotensin converting enzyme 2 (ACE2) receptor, through the RBD, the S2 domain plays a crucial role in the membrane fusion process[8]. The S glycoprotein is first synthetized in an inactive form and, only after priming by host proteases, it promotes the infection of susceptible cells by binding to ACE2 receptor[9]. Therefore, the S glycoprotein undergoes drastic structural changes during the cell entry process[10]. These changes, as well as its high glycosylation degree, may be affecting its immunogenicity. Studies performed with several surface proteins that are analogous to the SARS-CoV-2 S (e.g., F protein of respiratory syncytial virus) showed that protein stabilization in its prefusion conformation maintained the exposed neutralizing epitopes and increased yield and immunogenicity[11]. In this line, Pallesen et al.[12] described that MERS-CoV S glycoprotein could be stabilized by two prolines (V1060P and L1061P). A similar strategy was applied to the S trimer of other coronaviruses, indicating that two proline substitutions (2P) between the heptad repeat region 1 (HR1) and the central helix could be a universal strategy for stabilizing coronavirus-derived S glycoproteins[12]. Accordingly, Wrapp et al. showed that the 2P strategy also stabilized the SARS-CoV-2 S in its prefusion conformation[13]. Thus, the S-2P glycoprotein has been used as an immunogen in several COVID-19 vaccines, including those developed by Pfizer/BioNTech, Moderna, and Janssen[3]. However, in the presence of the 2P mutations, the S glycoprotein remains some structural motility and the RBD can be found in an up (open) or down (close) conformation[13].

Here, we explored alternative stabilization approaches of the SARS-CoV-2 S glycoprotein. Through extensive computational modeling, we identified a mutation (V987H) that improved the production of the recombinant S trimer and the exposure of the RBD. Even though both V987H Spike (S-V987H) and S-2P variants showed equivalent immunogenicity, S-V987H showed a better prophylactic activity than S-2P in K18-hACE2 mice and golden Syrian hamsters (GHS) against SARS-CoV-2 D614G, B.1.351 (Beta), and Omicron BQ1.1 variants.

## Results

### Identification of S glycoprotein mutations that constrain the motility of RBD

The native SARS-CoV-2 S trimer possess some structural flexibility that affects its stability and immunogenicity[10,13,14]. In addition to pre- and post-fusion S conformations, each RBD displays a dynamic equilibrium between open (up) and closed (down) configurations. In this regard, we aimed to design S variants with a preference for adopting the closed state, and thus, showing limited opening motion and RBD exposure. To this end, we envisioned a computational pipeline involving the three-dimensional modeling of all possible single mutations for both open and closed states, followed by the estimation of changes in their Gibbs free energy ($\Delta\Delta G$) (Fig. 1a). We focused on all single mutations showing a strong predicted preference for the closed-state ($\Delta\Delta G < -1$ kcal/mol) and among them, only those that clearly generated well-defined interactions (hydrogen bonds, ionic interactions of filling hydrophobic pockets) between the RBDs of the trimer were screened. We selected a total of 11 single mutations (A372W, K386R, G416R, D420R, D420Y, D427I, L517Y, S982F, D985L, V987H, and

V987W) (Fig. 1a). We also included one double mutation (A372W-D420R) referred as 2 M, and a combined quintuple mutation (D198F-G232L-A372W-N394Q-D420R) named 5 M. Locations of the selected mutations are represented in Fig. 1b. Recombinant mutant proteins were expressed by transient transfection in Expi293F cells, and their production was evaluated by ELISA (Fig. 1c). Most variants displayed a substantial decrease in production when compared to the S-2P trimer (Fig. 1c). Then, we analyzed the exposure of the RBD by ELISA using a Fc fusion protein containing the extracellular portion of the human ACE2 receptor fused to the human IgG1-Fc domain. The results confirmed that most of the variants were in fact promoting a closed trimer conformation (Fig. 1d) as it was predicted by our in silico pipeline. Moreover, variants associated with a reduced exposure of the RBD also resulted in a very low production. In contrast, the V987H mutation promoted the exposure of the RBD (Fig. 1d) and showed higher production than the S-2P protein (2.5-fold). Thus, our results suggest that the RBD exposure may be associated with S production levels.

### S-V987H trimer vaccination protects K18-hACE2 mice from SARS-CoV-2 D614G infection-associated disease

It has been described that K986P and V987P mutations stabilize and increase the expression and immunogenicity of the Spike glycoprotein[13,14]. Since the V987H mutation improved Spike trimer production and RBD exposure, we evaluated whether it could impact the Spike antigenicity in vivo. Thus, we compared the immunogenicity and protective capacity of the recombinant S-2P, S-V987H and RBD (Fig. 2a) after SARS-CoV-2 D614G challenge in K18-hACE2 mice (Fig. 2b).

Forty-five K18-hACE2 mice were immunized using a DNA prime-protein boost immunization strategy since it has been shown that this approach can generate T and B cell responses[15] (Fig. 2b). In addition to S-V987H ($n = 14$), we determined the immunogenicity of S-2P ($n = 16$), and a recombinant monomeric RBD protein ($n = 15$). Mice were first immunized by DNA electroporation with 40 µg of plasmid. Two weeks later, animals were boosted with the corresponding purified recombinant protein (15µg) formulated with aluminum phosphate as adjuvant. Prior to challenge, four mice from S-2P and control groups, two mice from S-V987H and three mice from RBD groups were euthanized to collect tissue samples. Then, 12 vaccinated mice for each group, and 16 unvaccinated controls were intranasally challenged with SARS-CoV-2 D614G (Fig. 2b). Four mice (two male/two female) from each group were euthanized on days 2, 4, and 7 (end of the experiment) post-challenge (Fig. 2b) to analyze the humoral immune responses, viral replication in target organs, and tissue damage. Mice that developed severe SARS-CoV-2 induced disease and/or showed a weight reduction higher than 20% of the initial weight were euthanized before the end of the experiment (day 7) as a humane endpoint. Four additional unvaccinated mice were used as uninfected controls.

The humoral responses elicited against the RBD (Fig. 2c), and the S protein (Supplementary Fig. 1a) were evaluated before each immunization (days −28 and −14) and viral challenge (day −2), and in the mice euthanized on days 2, 4 and 7 after infection. At all-time points, mice immunized with S-V987H and S-2P showed similar levels of anti-RBD and anti-Spike IgG antibodies, which were greater than those observed in RBD vaccinated animals ($p < 0.001$, Conover-Iman test) (Fig. 2c and Supplementary Fig. 1a–c). For simplification purposes, challenged animals were grouped as a "post-challenge" group (Fig. 2c). The levels of anti-RBD and anti-Spike IgG antibodies in the S-2P and S-V987H groups increased after boost and viral challenge ($p < 0.05$, Conover-Iman test) (Fig. 2c and Supplementary Fig. 1a), while mice immunized with RBD only presented increased levels of these antibodies after viral challenge ($p < 0.05$, Conover-Iman test), indicating that infection may further boost humoral responses in vaccinated mice (Fig. 2c and Supplementary Fig. 1a–c).

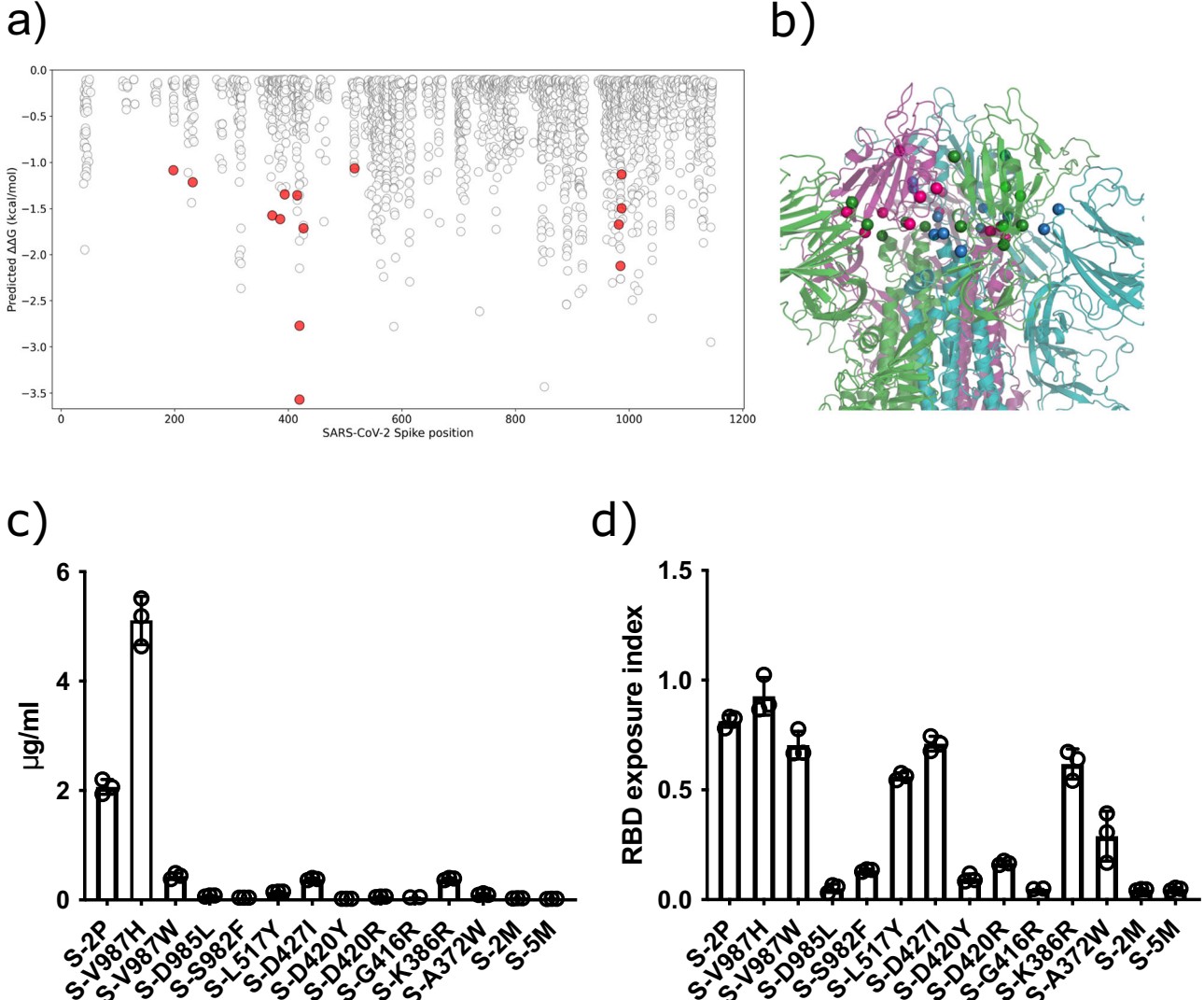

**Fig. 1 | Selection of mutations that stabilized the S glycoprotein in a closed conformation.** An in silico saturation mutagenesis study was performed for selecting mutations that could stabilize the S glycoprotein in a closed conformation. The production of selected variants and the exposure of the RBD was evaluated by ELISA. **a** Saturation mutagenesis of the SARS-CoV-2 S glycoprotein. Mutations selected for experimental characterization (with a favorable predicted $\Delta\Delta G < -1$ kcal/mol and showing stabilizing interactions between the RBDs in the closed conformation) are colored depending on the mutant residue. Labeled mutations were selected for experimental characterization. **b** Three-dimensional location of the selected mutations (displayed as dots and colored by their belonging monomer) in the closed state of the SARS-CoV-2 S glycoprotein (PDB:6VXX). Mutations are in multiple domains of the S glycoprotein, including NTD, CDT1, CDT2 and HR1-CH region. **c** Levels of recombinant proteins in a 5-day cell culture supernatant of transiently transfected Expi293F cells. Mean plus standard deviation of three experiments are shown. **d** RBD exposure index in selected recombinant proteins. Data are shown as ratio between RBD binding and total protein. Mean plus standard deviation of three experiments are shown. Source data from (**c**) and (**d**) are provided as a Source Data Fig. 1).

In addition, we evaluated level of NAbs against the Wuhan-Hu−1 (WH1) strain and Beta (B.1.351) variant after SARS-CoV-2 challenge. Both S-2P or S-V987H mice groups developed equivalent titers of NAbs against SARS-CoV-2 WH1, but significantly higher than those measured in the RBD immunized group (Fig. 2d $p < 0.05$, Conover-Iman test). However, only S-2P and S-V987H vaccinated mice had systemic neutralizing activity against the Beta VoC prior to challenge (Fig. 2e). Despite this, the titers of NAbs against WH1 strain were higher than those targeting the Beta VoC (Supplementary Fig. 1d).

The progressive increase in the levels of total IgG antibodies and/ or NAbs observed in both RBD and control-challenged groups after infection (Fig. 2c, d) also supports the idea of a boosting of the humoral response after virus challenge. To better characterize the humoral response against SARS-CoV-2 Omicron BA4/5 and BQ1.1 variants, and due to the lack of serum samples from K18-hACE2 mice, we immunized C57BL/6 mice following the DNA-prime/protein-boost strategy used with K18-hACE2 (Supplementary Fig. 1e). Serum samples were collected 14 days after protein boost and the Omicron BA4/5 and BQ1.1 neutralizing activity determined in vitro. Low levels of BQ1.1 neutralizing activity were detected in seven out of ten S-2P or S-V987H immunized mice (Supplementary Fig. 1f). However, only two serum samples from S-2P and three from S-V987H immunized mice neutralized BA4/5 (Supplementary Fig. 1g). No neutralizing activity was observed in RBD vaccinated animals for any of the two Omicron variants. In addition, we used splenocytes from these C57BL/6 immunized mice to assay the development of T cells responses. Spike-specific interferon-γ producing T cells were detected by ELISpot in mice vaccinated with S trimers. RBD vaccination induced a poor T cell responses in these animals (Supplementary Fig. 1h).

To assess the ability of each immunogen to prevent SARS-CoV-2 infection-associated disease, we measured weight evolution in all

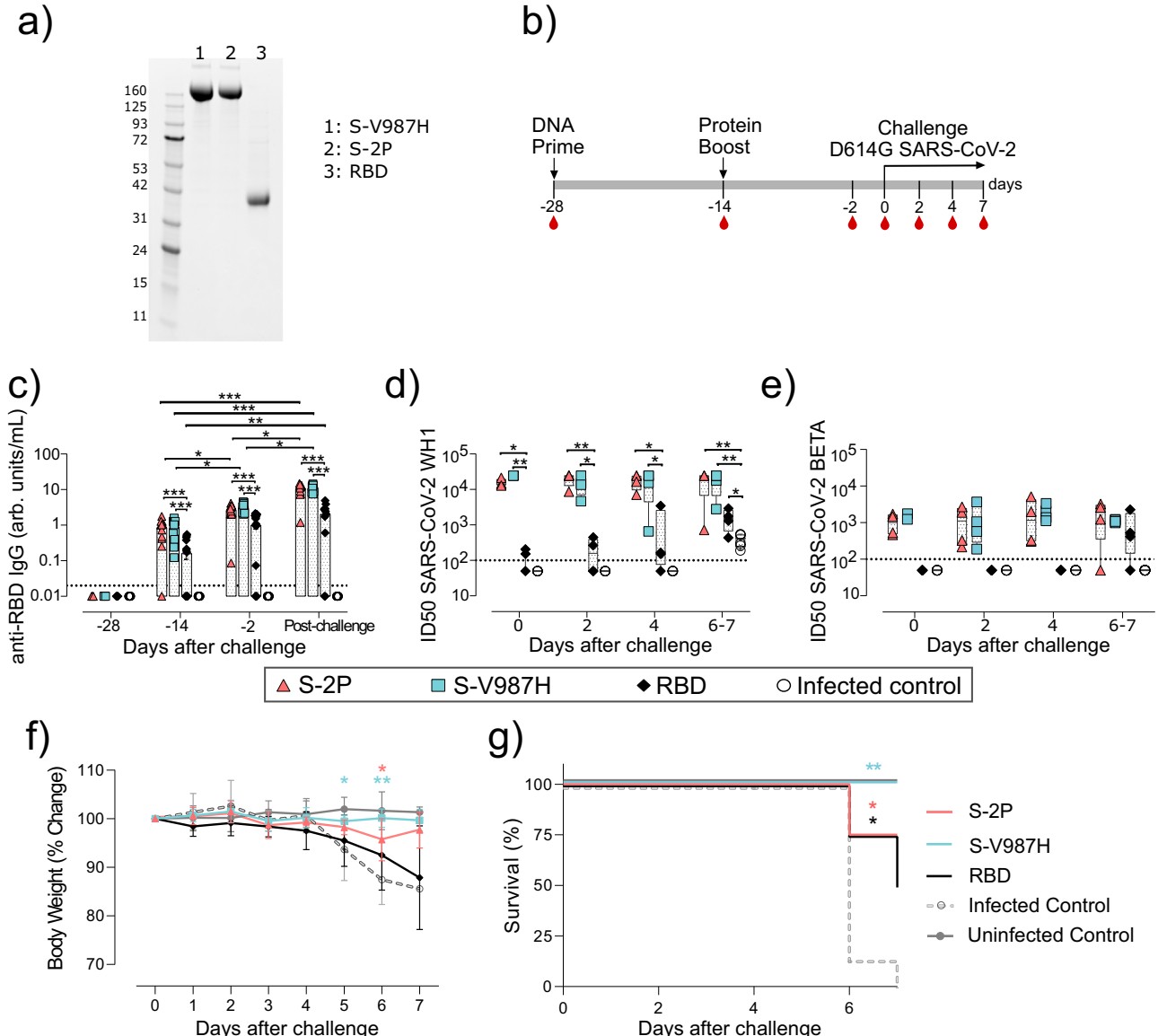

**Fig. 2 | Prophylactic activity of S-V987H immunization and analysis of the humoral response elicited in vaccinated K18-hACE2 mice challenged with the SARS-CoV-2 D614G variant.** K18-hACE2 mice were immunized following a DNA/protein prime/boost strategy with S-V987H, S-2P or RBD, and challenged with SARS-CoV-2 D614G. The humoral response, weight changes and survival of mice were evaluated after immunization and/or viral challenge. **a** Purified S-V987H, S-2P and RBD were analyzed by SDS-PAGE and Coomassie G-250 staining. Data are representative of six (S-V987H and S-2P) and two (RBD) independent purifications. **b** Overview of vaccination strategy and infection timeline. Blood drops indicate collection of biological samples. **c** Kinetics of anti-RBD IgG in serum samples expressed as arbitrary units (arb. units) per mL. Red triangles: S-2P group ($n = 16$). Blue squares: S-V987H group ($n = 14$). Black diamonds: RBD group ($n = 15$). White circles: unvaccinated and challenged mice (infected control) ($n = 19$). Groups in each time point were analyzed using two-tailed Conover Iman test with multiple comparison correction by FDR. Differences among animals within a particular group along time were analyzed using the two-tailed Friedman test corrected for multiple comparison using FDR. Mean plus standard errors of the means (SEM) are shown. **d** Levels of NAbs against SARS-CoV-2 WH1 variant after viral challenge. **e** Levels of NAbs

against SARS-CoV-2 B.1.351 (Beta) variant after viral challenge. Neutralization data were analyzed as indicated in (**c**). Four mice per timepoint were analyzed in (**d**) and (**e**), with two exceptions: 1) day 0: S-V987H = 2, RBD = 3, and Control = 3; and 2) days 6–7: infected control = 8. **f** Percentage of weight variation in SARS-CoV-2 D614G-infected K18-hACE2 mice over time. Statistical analysis was performed comparing each vaccinated group with the unvaccinated group using Kruskal-Wallis test corrected by FDR (two-tailed). Mean and standard deviation (SD) are shown. S-2P: $n = 12$ on days 0–2, $n = 8$ on days 3–4, $n = 4$ on days 5–6, and $n = 3$ on day 7. S-V987H and RBD: $n = 12$ on days 0–2, $n = 8$ on days 3–4, and n = 4 on days 5–7. Unvaccinated-challenged group (infected control): $n = 16$ on days 0–2, $n = 12$ on days 3–4, $n = 8$ on days 5–6, and $n = 1$ on day 7. Four animals per timepoint were analyzed in the uninfected control group. **g** Kaplan-Meier plot showing the survival rate during the course of the experiment. S-2P, S-V987 and RBD groups $n = 4$, Unvaccinated-unchallenged group (infected control) $n = 8$. Uninfected control $n = 4$. Statistical analysis was performed against unvaccinated group using two-sided Mantel-Cox test. *$p < 0.05$, **$p < 0.01$, ***$p < 0.001$. Data represented in (**c**–**g**) correspond to one experiment. Source data are provided as a Source Data Fig. 2.

groups of K18-hACE2 mice after SARS-CoV-2 D614G challenge, as an indicator of disease progression in this model[16]. We identified weight reduction on day 5 post-infection in mice belonging to infected-control and RBD groups, which is opposed to S-2P and S-V987H vaccinated groups ($p < 0.05$, Kruskal-Wallis test corrected by FDR)

(Fig. 2f). On day 6 and 7, all animals from the inoculated-control group ($n = 4$), one out of four S-2P vaccinated mice, and two out of four RBD immunized mice had to be euthanized due to disease development (Fig. 2g; $p < 0.05$ compared to control-infected group, Log-rank test) Mice from the S-V987H group did not show clinical signs of disease

during the entire experimental procedure ($p < 0.01$ compared to control-infected group, Log-rank test) (Fig. 2f, g).

In addition to the clinical course, we measured the levels of viral replication by RT-qPCR in four samples (Fig. 3a): oropharyngeal swab, nasal turbinate, lung, and brain. Although the levels of genomic RNA (gRNA) in oropharyngeal swabs decreased in all groups over time, only S-V987H vaccinated mice became undetectable on day 7. On the contrary, S-2P and RBD vaccinated mice remained positive (Fig. 3a, $p < 0.05$ Peto & Peto Left-censored k sample test). Similar results were observed in nasal turbinate (Fig. 3a), where gRNA was lower in S-2P and S-V987H groups compared to control-infected mice or the RBD group on days 6–7 after viral challenge, and the S-V987H group showed the lowest values by day 7. Interestingly, low levels of gRNA were detected in lung and brain from S-V987H immunized mice (lung $p = 0.055$, brain $p < 0.05$, Peto & Peto Left-censored k sample test) (Fig. 3a), whereas a progressive increase was observed in brains from the infected-control and RBD groups, and in the only S-2P vaccinated mouse that developed disease (Fig. 3a). Conversely, subgenomic RNA (sgRNA), which indicates active viral replication, was not detected at any time point in lung or brain of mice immunized with any of the S trimers, or in oropharyngeal swabs on day 4 and 7, except for the single S-2P mouse that developed illness (Supplementary Fig. 2). The presence of infectious viruses (IVs) was investigated monitoring the cytopathic effect on Vero E6 cells. Only samples with a gRNA Ct value < 30 were analyzed. IVs were hardly detected in oropharyngeal swabs and in the remaining tissues of trimer vaccinated mice (Fig. 3b). However, they were consistently detected in tissues from controls and, to a lesser extent, from RBD immunized mice. In addition, high titers of IVs were detected in lung and brain of the S-2P immunize mouse that reached humane end point (Fig. 3b).

Active viral replication was also analyzed measuring nucleoprotein (NP) levels in tissues by immunohistochemistry (IHC). NP was detected in lung and brain of both control and RBD groups; and in one animal from S-2P group that developed the disease, but not in S-V987H or disease-free S-2P-vaccinated mice after challenge (Fig. 3c, $p < 0.05$ asymptotic generalized Pearson Chi-Squared test corrected for multiple comparison using FDR). Low IHC scores were observed in nasal turbinates on days 2 and 4 with no major differences among study groups (Fig. 3c). Tissue damage was in line with the levels of viral antigens detected by IHC (Fig. 3d). No tissue damage was observed in lung or brain of mice vaccinated with S-2P and S-V987H, except for the S-2P mouse that became sick (Fig. 3d). A low lesion score was recorded at early time points after challenge in nasal turbinate of all infected mice (Fig. 3d).

Overall, the immunogenicity of both S-2P and S-V987H trimers was equivalent in K18-hACE2 and C57BL/6 mice, and greater than the produced by the monomeric RBD immunogen. However, S-V987H vaccination improved mice protection against SARS-CoV-2 D614G variant over the S-2P immunogen, since all mice in S-V987H group were disease-free and showed a faster viral clearance in tissues.

### S-V987H trimer vaccination protects golden Syrian hamsters from SARS-CoV-2 infection-associated disease

To confirm the results obtained in the transgenic mouse model, we performed a second experiment using golden Syrian hamster (GSH). Similar to K18-hACE2 mice, GSH were immunized using a prime-boost strategy, and intranasally challenged with SARS-CoV-2 D614G (Fig. 4a). Animals were monitored until day 7 post-inoculation, since it has been described that animals start spontaneously recovering around a week after viral infection[17,18].

The magnitude of the humoral responses elicited against the S and the RBD by both S-2P and S-V987H trimers was similar and greater than those elicited by the RBD immunogen (Fig. 4b and Supplementary Fig. 3a). The levels of anti-RBD and anti-S IgG antibodies increased after each immunization and after viral challenge ($p < 0.05$, Friedman test)

(Fig. 4b and Supplementary Fig. 3a), confirming the results obtained in K18-hACE2 mice. However, unlike mice, infected-control GSHs rapidly developed an anti-S humoral response after challenge, showing similar levels of anti-S and anti-RBD antibodies on day 7 to those observed in animals immunized with the RBD protein (Fig. 4b and Supplementary Fig. 3b). When the neutralizing activity of serum samples was analyzed, we observed that GSHs immunized with S-2P or S-V987H proteins neutralized the WH1 variant and, to a lesser extent, the Beta VoC (Fig. 4c, d). The neutralizing activity against WH1 increased overtime after challenge in all study groups ($p < 0.05$, Conover-Iman test). Neutralization of WH1 was also detected in sera from infected control animals by day 4 after challenge, and their titers rapidly increased, becoming similar to the ones observed in S-V987H and RBD groups, and higher than those observed in S-2P vaccinated animals by day 7 ($p < 0.05$, Conover-Iman test). Intriguingly, despite all groups showed similar titers of NAbs on day 7 after challenge, the levels of anti-RBD and anti-S binding antibodies (Fig. 4b and Supplementary Fig. 3a, b) were higher in the S-2P and S-V987H immunized groups than in infected-controls GSH. These results support that SARS-CoV-2 infection induced a rapid humoral response against SARS-CoV-2 in GSH that may be qualitatively different to the one elicited by immunization.

We then evaluated the clinical course after challenge. Animals in both control and RBD groups showed a progressive weight reduction until day 7 (end of the experiment) indicative of disease progression (% of weight in infected controls = $87.3 \pm 3.1$; RBD group = $84.4 \pm 1.4$) (Fig. 4e). Such weight loss was not observed in S-2P ($98.9 \pm 1.3$) or S-V987H ($98.76 \pm 2.4$) vaccinated GSH ($p < 0.05$ Kruskal-Wallis test corrected by FDR). Thus, both S trimers generated equivalent protection from disease development in vaccinated GSH (Fig. 4e).

The presence of SARS-CoV-2 was determined by RT-qPCR in oropharyngeal swabs and respiratory tissue samples (nasal turbinate and lung). Brain was not evaluated in GSHs since SARS-CoV-2 does not affect the brain in this animal model[19]. The levels of gRNA decreased over time in all analyzed samples from both S-trimers immunized GSH. In addition, we detected lower gRNA levels in nasal turbinate (day 7) and in lung (days 2, 4 and 7) of both S-2P and S-V987H groups compared to the RBD and infected-control groups (Fig. 5a; $p < 0.01$, Peto & Peto Left-censored k sample test). No major differences were detected in the levels of gRNA in oropharyngeal swabs among the study groups (Fig. 5a). While we did not observe significant differences among groups when sgRNA was analyzed in nasal turbinate or oropharyngeal swabs, we detected lower sgRNA levels in the lungs of S-V987H and S-2P groups compared to RBD and infected-control groups on days 2 and 4 after challenge (Supplementary Fig. 3c).

IVs were hardly detected in oropharyngeal swabs regardless the time point analyzed (Fig. 5b). Consistently with gRNA data, a progressive reduction in IV levels were observed in all study groups in nasal turbinate and lung. Importantly, whereas all vaccinated GSHs showed lower levels of IVs than control animals in lung on day 2 post-inoculation, IVs were not detected in lung samples from the S-V987H group on day 4, indicating a faster viral clearance in this group when compared with S-2P, RBD or unvaccinated controls (Fig. 5b). In line with this observation, S-V987H immunized GSH showed the lowest levels of IVs in nasal turbinate on day 4 post SARS-CoV-2 inoculation (Fig. 5b).

These results were also aligned with the levels of NP detection by IHC (Fig. 5c). No major differences in NP levels were observed among study groups at any time points in nasal turbinate, becoming undetectable by day 7 (Fig. 5c) ($p < 0.05$, Asymptotic Generalized Pearson Chi-Squared Test). However, lower NP levels were detected in lungs of both S-2P and S-V987H vaccinated groups when compared with RBD and infected controls on days 2, 4, and 7. Interestingly, NP was not detected in lungs on day 7 in S-2P and S-V987H groups (Fig. 5c). All study groups showed a similar lesion degree in nasal turbinate, which decreased by day 7 after challenge ($p < 0.05$). By contrast, a lower

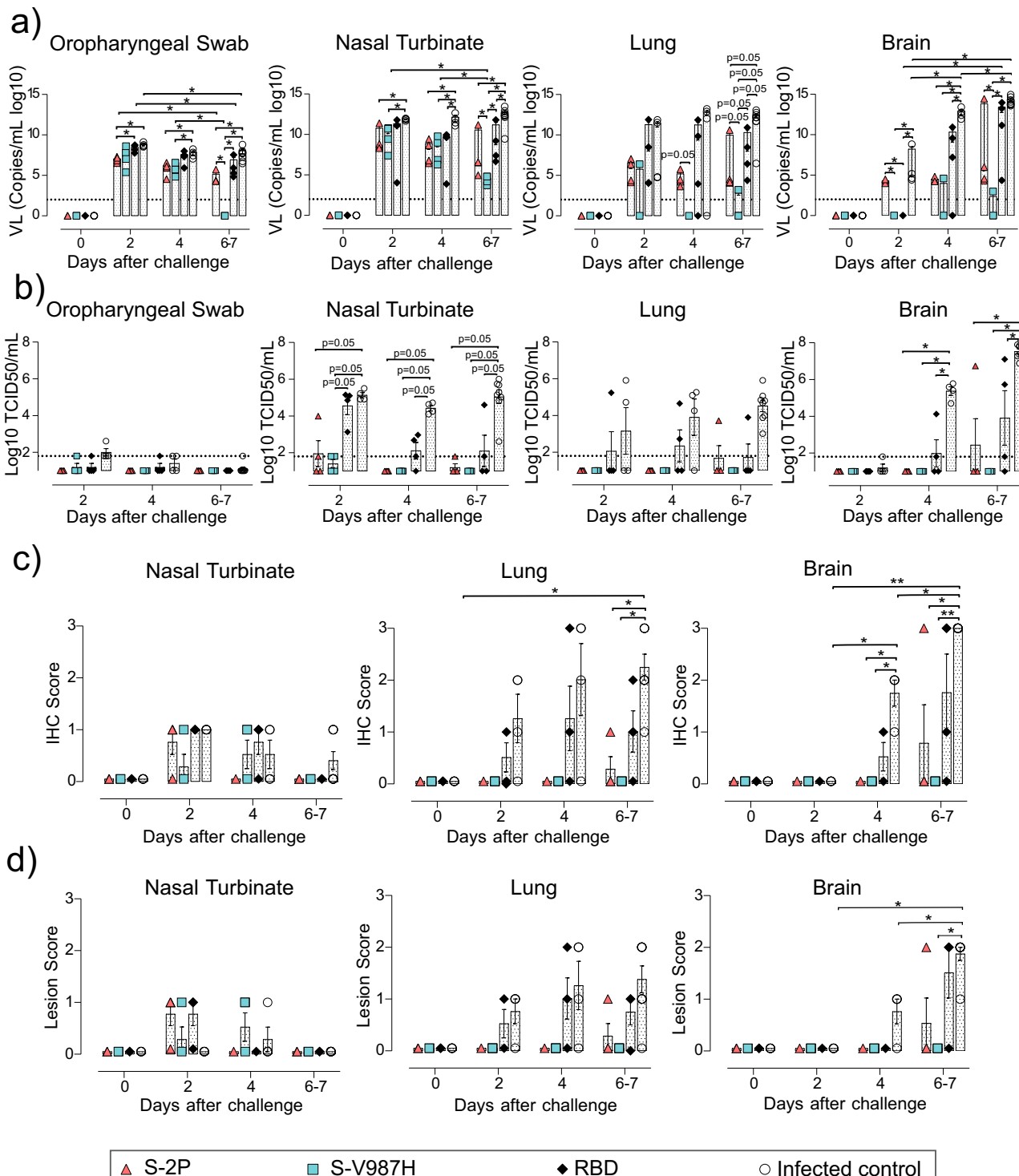

**Fig. 3 | Viral load and histopathological analysis of tissue samples from SARS-CoV-2 D614G infected K18-hACE2 mice after vaccination.** SARS-CoV-2 viral load was analyzed in oropharyngeal swabs, and samples from nasal turbinate, lung, and brain of K18-hACE2 mice upon challenge. Virus distribution and tissue damage were analyzed by histopathology. Four mice per timepoint were analyzed, with two exceptions: 1) day 0: S-V987H = 2, RBD = 3, and Infected control = 3; and 2) days 6–7: Infected control = 8 **a** Levels of SARS-CoV-2 gRNA (expressed as logarithmic of copies/mL) in oropharyngeal swabs, nasal turbinate, lung, and brain during infection. Dot line indicates limit of detection (100 copies/mL). Differences between animals were analyzed using two-sided Peto & Peto left-censored k sample test, correcting by FDR. **b** Titer of infectious virions (IVs) in oropharyngeal swabs, nasal turbinate, lung, and brain during infection. Results are shown as Log10 of Median

Tissue Culture Infectious Dose per mL (TCID50/mL). Differences between groups were analyzed as indicated in (**a**). **c** Detection of SARS-CoV-2 nucleocapsid protein in brain, lung, and nasal turbinate by immunohistochemistry. Staining score: (0) no, (1) low, (2) moderate, and (3) high amount of viral antigen. **d** Histopathological analysis of nasal turbinate, lung and brain by hematoxylin and eosin staining. Lesion score: (0) no, (1) mild, (2) moderate, and (3) severe lesion. Differences between groups were analyzed using two-sided Asymptotic Generalized Pearson Chi-Squared test with FDR correction. *$p < 0.05$, **$p < 0.01$. P values proximal to statistical significance are shown as numbers. Mean plus standard error of the mean (SEM) are shown. Data represented in Fig. 3 correspond to one experiment. Source data are provided as a Source Data Fig. 3.

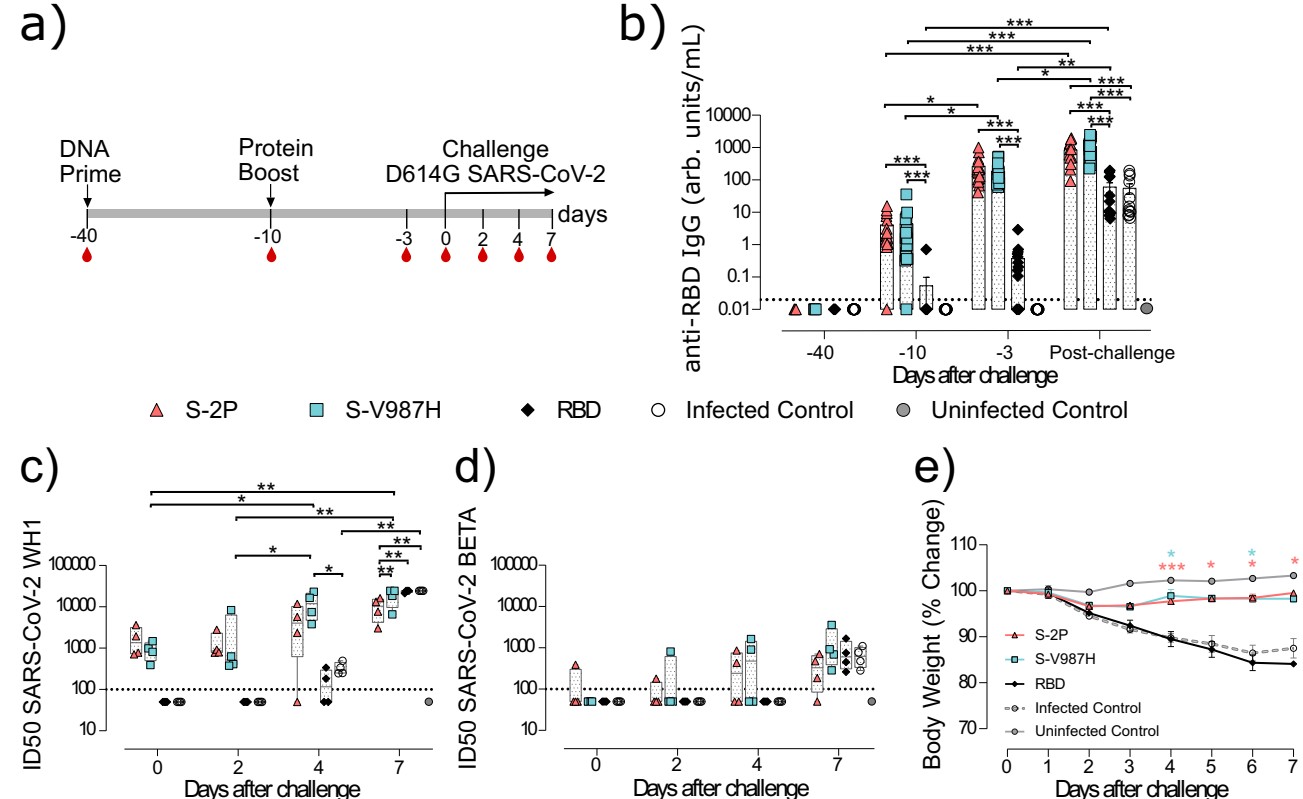

**Fig. 4 | Prophylactic activity of S-V987H immunization and analysis of the humoral response elicited in vaccinated golden Syrian hamsters challenged with the SARS-CoV-2 D614G variant.** GSH were immunized following a prime/ boost strategy with S-V987H, S-2P or RBD, and challenged with SARS-CoV-2 D614G. The humoral response and weight changes of GSH were evaluated after immunization and/or viral challenge. **a** Outline of vaccination strategy and infection timeline. Blood drops indicate collection of biological samples. **b** Kinetics of anti-RBD antibodies in serum samples expressed as arbitrary units (arb. units). Red triangles: S-2P group ($n = 16$). Blue squares: S-V987H group ($n = 16$). Black diamonds: RBD ($n = 16$). White circles: unvaccinated and challenged GSH (infected control) ($n = 16$). Gray circles: unvaccinated and uninfected GSH (uninfected control) ($n = 4$). Groups in each time point were analyzed using two-sided Conover-Iman test with multiple comparison correction by FDR. Differences among animals

within a particular group along time were analyzed using the two-sided Friedman test with FDR correction. Mean plus standard error of the mean (SEM) are shown. **c** Sera neutralizing activity against SARS-CoV-2 WH1 and **d** SARS-CoV-2 B.1.351 (Beta) variants after viral challenge ($n = 4$ per timepoint and group). Neutralization data were analyzed as indicated in (**b**). **e** Percentage of weight variation in SARS-CoV-2 D614G-infected GSH over time. Statistical analysis was performed against the unvaccinated group using two-sided Kruskal-Wallis test correcting by FDR. Animals in S-2P, S-V987H and infected control were distributed as follows: days 0, 1, and 2 $n = 12$; days 3 and 4 $n = 8$; days 5, 6, and 7 $n = 4$. Uninfected control group $n = 4$ in each timepoint. Mean plus standard error of the mean (SEM) are shown *$p < 0.05$, **$p < 0.01$, ***$p < 0.001$. Data represented in Fig. 4 correspond to one experiment. Source data are provided as a Source Data Fig. 4.

tissue damage was observed in lung from S-V987H (on days 4 and 7) and in S-2P (on day 7) groups compared to RBD and infected control groups ($p < 0.05$) (Fig. 5d).

Overall, our results showed that the immunogenicity and protective efficacy of both S-2P and S-V987H trimers are equivalent in GSHs, and higher than the one conferred by RBD vaccination.

### S-V987H trimer vaccination protects K18-hACE2 mice from the SARS-CoV-2 Beta-variant challenge

From the beginning of the COVID-19 pandemic, several SARS-CoV-2 VoC have emerged. These VoC have shown different transmissibility, pathogenic potential and resistance to antibodies previously elicited by vaccination or natural infection[20]. The results described above have shown that S-V987H-vaccinated animals were protected from COVID-19 development after SARS-CoV-2 D614G strain challenge. Additionally, vaccinated animals showed low sera neutralizing activity against the SARS-CoV-2 Beta variant. Since the Beta VoC is one of the most resistant to antibodies elicited by natural infection and the currently available vaccines[21], and also induces severe disease in K18-hACE2 mice[22], we evaluated whether the immune responses induced by S-V987H could protect against disease development after challenge with the SARS-CoV-2 Beta variant. Thus, we immunized twenty-one

K18-hACE2 mice with S-V987H or S-2P, using AddaVax as adjuvant in this homologous prime-boost experiment (Fig. 6a). AddaVax is a MF59-like adjuvant that induces bot cellular and antibody responses[23]. Unvaccinated mice were used as negative ($n = 10$) and positive ($n = 16$) controls of infection. Two weeks after receiving the protein boost, mice were challenged with the SARS-CoV-2 Beta variant (Fig. 6a). Six mice from each challenged group were euthanized on days 3 ($n = 6$) and 6 ($n = 6$) after infection. The remaining animals were euthanized on day 14 after challenge, excepting those mice that developed severe disease after day 3 (10 in the infected-control group and one in the S-2P group) that were euthanized before day 14 following the humane endpoints of the protocol and analyzed separately. Of note, both S-2P and S-V987H recombinant proteins induced similar levels of IgG antibodies against the S and the RBD, which increased after each boost and after viral challenge ($p < 0.05$, Conover-Iman test) (Fig.6b and Supplementary Fig. 4a, b). Interestingly, three days after challenge, S-V987H immunized mice showed higher sera neutralizing activity against the WH1 ($n = 6$; $15376 \pm 9203$) (Fig. 6c), and the Delta VoC ($n = 6$; $7750 \pm 8403$) (Fig. 6d) than mice immunized with the S-2P ($n = 6$; WH1: $2913 \pm 3524$; Delta: $1505 \pm 4773$) (WH1: $p < 0.01$; Delta: $p = 0.055$; Conover-Iman test). Neutralizing activity against the Beta VoC increased after challenge ($p < 0.05$, Conover-Iman test) (Fig. 6e). In

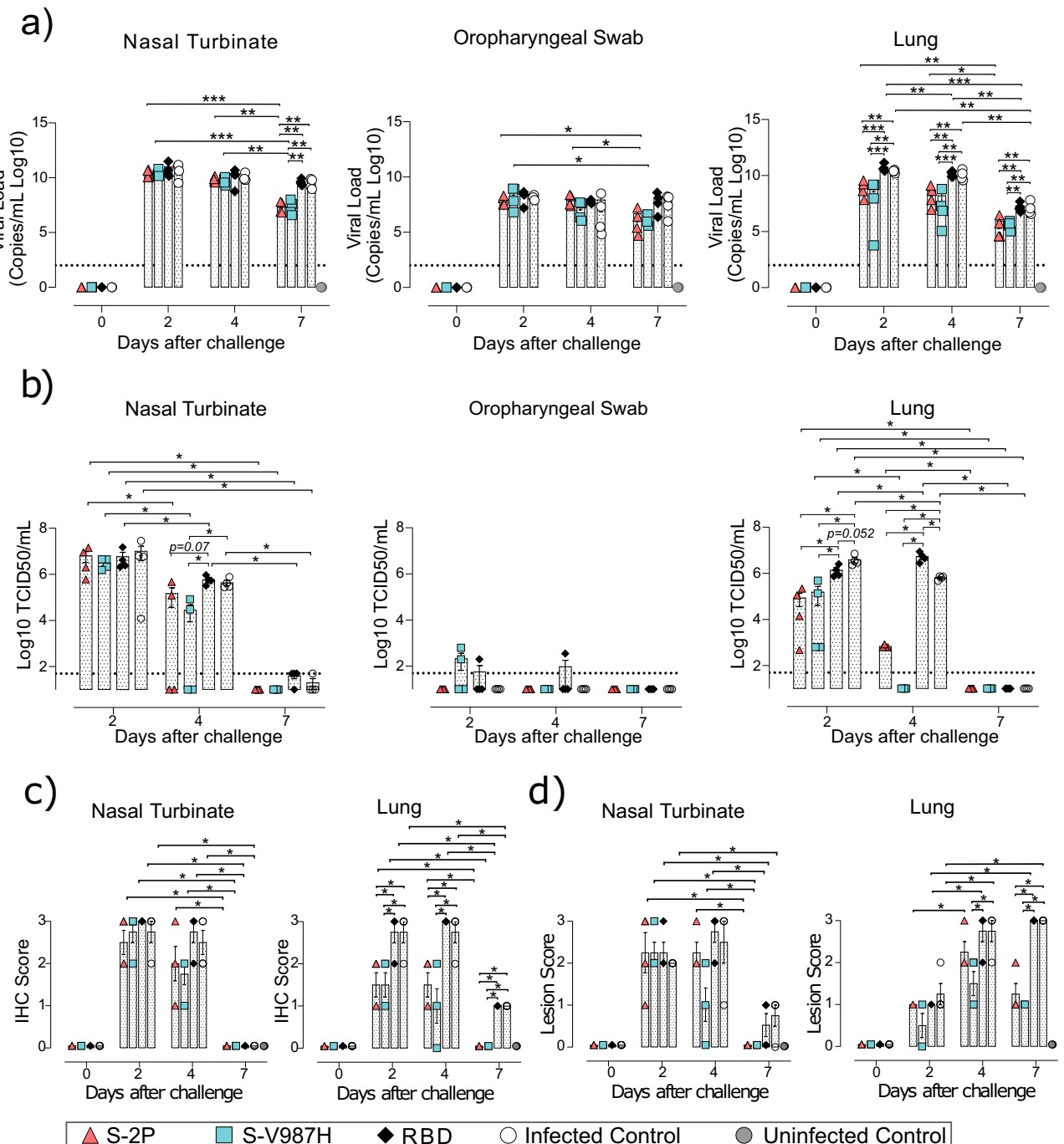

**Fig. 5 | Viral load and histopathology analysis of tissue samples from SARS-CoV-2 D614G infected golden Syrian hamsters after vaccination.** SARS-CoV-2 viral load was analyzed in oropharyngeal swabs, and samples from nasal turbinate and lung of infected GSHs. Virus distribution and tissue damage was analyzed by histopathology. S-2P, S-V987H and infected control *n* = 4 per group and timepoint. Uninfected control *n* = 4 on day 7. **a** Levels of SARS-CoV-2 gRNA, expressed as Log10 copies/mL, in oropharyngeal swabs, nasal turbinate and lung during infection. Dot line indicates limit of detection (100 copies/mL). Differences between groups were analyzed using two-sided Peto & Peto left-censored k sample test with FDR correction. **b** Titer of infectious virions determined in samples from nasal turbinate, oropharyngeal swab, and lung. Data are shown as Log10 of Median Tissue Culture Inhibition Dose per mL (TCID50/mL). Mean plus standard error of the mean (SEM) are shown. **c** Detection of SARS-CoV-2 nucleocapsid protein in lung and nasal turbinate by immunohistochemistry. Staining score: (0) no, (1) low, (2) moderate, and (3) high amount of viral antigen. **d** Histopathologic analysis of nasal turbinate and lung by hematoxylin and eosin staining. Lesion score: (0) no, (1) mild, (2) moderate, and (3) severe lesion. Differences between groups were analyzed by the two-sided Asymptotic Generalized Pearson Chi-Squared test corrected using FDR. Mean plus standard errors of the means (SEM) are shown. *\*p* < 0.05, \*\**p* < 0.01, \*\*\**p* < 0.001. Data represented in Fig. 5 correspond to one experiment. Source data are provided as a Source Data Fig. 5.

addition, we identified an increasing trend in sera neutralizing activity against Omicron BA.1 over time (p = 0.055, Conover-Iman test) (Fig. 6f). These differences suggest that the humoral responses elicited after S-2P or S-V987H immunization evolved after challenge with the SARS-CoV-2 Beta variant, increasing neutralizing activity against Beta and Omicron BA.1 VoC, as well as against Delta in the case of the S-2P group. Interestingly, neutralizing activity against the Beta VoC was detected in control-infected mice at clinical endpoint (Fig. 6e) with

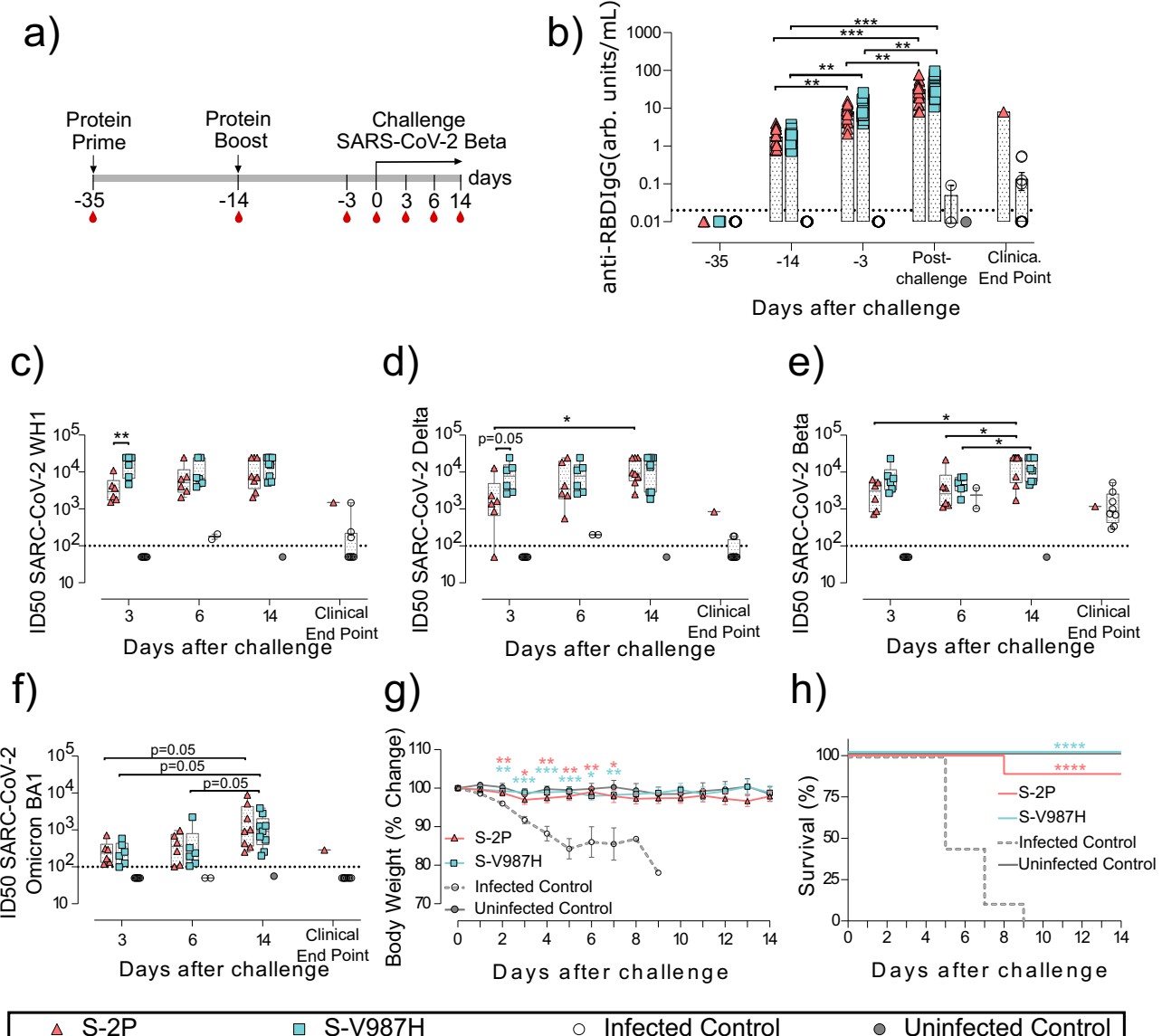

**Fig. 6 | Prophylactic activity of S-V987H immunization and analysis of the humoral response elicited in vaccinated K18-hACE2 mice challenged with the SARS-CoV-2 B.1.351 (Beta) variant.** K18-hACE2 mice were immunized twice with S-V987H or S-2P, adjuvanted with AddaVax. Then, mice were challenged with the SARS-CoV-2 B.1.351 VoC. The humoral response, weight changes and mice survival were evaluated after immunization and/or viral challenge. **a** Overview of vaccine strategy and infection timeline. Blood drops indicate collection of biological samples. **b** Kinetics of anti-RBD antibodies in serum samples expressed as arbitrary units (arb. units). Red triangles: S-2P group ($n = 21$ on days −35, −14 and −3; $n = 20$ post-challenge, $n = 1$ clinical endpoints). Blue squares: S-V987H group ($n = 21$ on all timepoints). White circles: unvaccinated-challenged mice (Infected control) ($n = 16$ on days −35, −14 and −3; $n = 8$ post-challenge, and $n = 8$ clinical endpoints). Gray circles: uninfected and unvaccinated mice (Uninfected control) ($n = 10$). Groups in each time point were analyzed using two-sided Conover-Iman test with multiple comparison correction by FDR. Differences among animals within a particular group along time were analyzed using two-sided Friedman test with FDR correction. Sera neutralizing activity against: **c** SARS-CoV-2 WH-1, **d** B.1.617.2 (Delta), **e** B.1.351 (Beta), and **f** B.1.1.529 (Omicron) variants after viral challenge. Neutralization data were analyzed as indicated in **b**. S-2P group: $n = 6$ on day 3, $n = 6$ on

day 6, $n = 8$ on day 14, $n = 1$ on clinical endpoints. S-V987H group: $n = 6$ on day 3, $n = 6$ on day 6, $n = 9$ on day 14. Unvaccinated-challenged (Infected control) mice: $n = 6$ on day 3, $n = 2$ on day 6, $n = 8$ on clinical endpoints. Uninfected and unvaccinated mice (Uninfected control): $n = 10$ on day 14. **g** Percentage of weight variation in SARS-CoV-2 B.1.351 infected K18-hACE2 mice over time. Statistical analysis was performed against the unvaccinated and challenged group using two-sided Kruskal-Wallis test with FDR correction. S-2P group: $n = 21$ on days 0–3, $n = 15$ on days 4–6, $n = 9$ on days 7–8, and $n = 8$ on days 9–14. S-V987H group: $n = 21$ on days 0–3, $n = 15$ on days 4–6, $n = 9$ on days 7-14. Unvaccinated-challenged mice: $n = 16$ on day 0–3, $n = 10$ on day 4–5, n = 5 on day 6, $n = 3$ on day 7, and $n = 1$ on days 8–9. Uninfected and unvaccinated mice: $n = 10$ on day 0–14 **h** Kaplan-Meier plot showing the percentage of SARS-CoV-2-infected animals that survive after challenge. S-2P and S-V987H groups $n = 9$, Unvaccinated-challenged mice $n = 10$, Uninfected and unvaccinated mice: $n = 10$; Statistical analysis was performed against unvaccinated group using two-sided Mantel-Cox test. *$p < 0.05$, **$p < 0.01$, ***$p < 0.001$, ****$p < 0.0001$. Mean plus standard errors of the means (SEM) are shown. Data represented in Fig. 6 correspond to one experiment. Source data are provided as a Source Data Fig. 6.

little or no cross-neutralization activity with other variants (Fig. 6c, d, f). No statistical differences in neutralizing activity were observed on days 6 and 14 between S-V987H and S-2P groups for any of four SARS-CoV-2 variants evaluated (Fig. 6c–f). To evaluate whether the

homologous prime/boost immunization protocol using AddaVax as adjuvant improved the neutralizing activity against most recently identified Omicron variants (BA4/5 and BQ1.1), we immunized C57BL/6 mice using the protocol described in Fig. 6a. Despite our immunogen

was based on SARS-CoV-2 WH1 sequence, neutralizing activity was detected against Omicron BQ1.1 and BA 4/5 variants in serum samples collected two weeks after the boosting dose (Supplementary Fig. 4c). In addition, we used splenocytes from these C57BL/6 immunized mice to quantified S-specific interferon-γ producing T cells by ELISpot. S-specific T cells responses were detected in both immunized groups (Supplementary Fig. 4d).

After SARS-CoV-2 Beta challenge, a reduction of body weight associated with disease progression was observed in K18-hACE2 mice from the infected control group starting on day 2 after challenge compared to mice vaccinated with S-2P and S-V987H (Fig. 6g) ($p < 0.05$, Kruskal-Wallis test corrected by FDR). Mice in both Spike-vaccinated groups maintained their weight until day 14 [percentage of weight: S-2P = 99 ± 4 ($n = 9$); S-V987H = 98 ± 5 ($n = 9$)]. Contrarily to the S-2P and infected control groups, no mice from the S-V987H group ($n = 9$) showed any clinical signs of disease (Fig. 6g, h) during the experiment (day 14) ($p < 0.001$, Long rank test).

The analysis of viral load in tissues by RT-qPCR showed that both S-2P and S-V987H vaccinated groups had a progressive decrease in gRNA levels in oropharyngeal swabs and lung over time (Fig. 7a) ($p < 0.05$; Peto & Peto Left-censored k sample test). Interestingly, the S-V987H group displayed lower viral loads in nasal turbinate than S-2P and infected control animals on day 3, and also in oropharyngeal swab compared to the infected controls (Fig. 7a) ($p < 0.05$). However, these differences were not maintained over time and both S-trimer immunized groups showed low but equivalent values of gRNA on day 14 in all analyzed tissues (Fig. 7a). In addition, these groups displayed lower viral load in lung and brain compared to the infected control group at day 3 after challenge (Fig. 7a, $p < 0.05$). The levels of sgRNA in these tissues were in line with the results of gRNA (Supplementary Fig. 4e).

We investigated the presence of IVs in tissue samples with a gRNA Ct<30 by a plaque assay on Vero E6 cells. IVs were hardly detected in oropharyngeal swabs and in the remaining tissues of trimer vaccinated mice. However, they were consistently detected in nasal turbinate and lung tissues (day 3), and in brain (day 6 and clinical end point) of SARS-CoV-2 challenged control mice (Fig. 7b).

Remarkably, NP was hardly detected in lung and brain from S-2P and S-V987H groups by IHC (Fig. 7c), which was in line with the low levels of gRNA detected in these animals. Despite that, S-2P vaccinated mice showed a higher lesion score in lung at day 14 than the S-V987H group ($p < 0.01$; Asymptotic Generalized Pearson Chi-Squared test) (Fig. 7d), indicating that these mice presented a severe lung damage. Interestingly, both Spike-based immunogens protected from viral dissemination to the brain (Fig. 7a,b,c, d).

To summarize, the immunogenicity of both S-2P and S-V987H trimers was similar in K18-hACE2 SARS-CoV-2 Beta-infected mice, although S-V987H promoted the development of higher serum neutralization. However, the underlying mechanisms that conferred the slight increase in protection observed in S-V987H vaccinated animals compared to the S-2P group needs further investigation.

### S-V987H trimer immunization protects K18-hACE2 mice from the SARS-CoV-2 Omicron BQ1.1-variant challenge

Previously, we have demonstrated that S-V987H immunized C57Bl/6 mice induced Omicron BA 4/5 and BQ1.1 neutralizing antibodies (Supplementary Fig. 1f, g, and Supplementary Fig. 4c). To evaluate the efficacy of S-V987H protecting against the newest Omicron variants, we performed an immunization and challenge experiment in K18-hACE2 mice using the SARS-CoV-2 Omicron BQ1.1 variant (Fig. 8a). We chose this variant because it has been reported as pathogenic in K18-hACE2 mice[24]. Sixty K18-hACE2 mice were randomly allocated in four different groups: S-2P ($n = 18$) and S-V987H ($n = 18$) vaccinated groups, challenged control group previously immunized only with adjuvant ($n = 18$) and an unvaccinated/non-challenged negative control group ($n = 6$). We used AddaVax as adjuvant. Mice were immunized with the

corresponding S-trimer and challenged following the schedule described in Fig. 8a. As expected, S-2P and S-V987H immunizations induced anti-RBD (Fig. 8b) and anti-S IgG antibodies (Supplementary Fig. 5a) that increased with each immunization and after viral challenge (Fig. 8b, and Supplementary Fig. 5a,b). SARS-CoV-2 BQ1.1 infection did not induce a significant weight reduction in any group, irrespectively of their vaccination status (Fig. 8c), and all mice survive until the end of the study (day 14). These results were in line with the fact that gRNA was hardly detected in brain samples from any study groups (Fig. 8d). In addition, we observed a decrease in viral load over time after challenge in oropharyngeal swabs, nasal turbinate, and lung, which was faster in immunized animals. Thus, gRNA was not detected in oropharyngeal swabs from S-trimer immunized animals as of day 7 post-challenge, and S-V987H immunized mice showed lower viral levels than control animals in nasal turbinate on day 3 and in lung on days 3 and 7 (Fig. 8d) ($p < 0.05$; Peto & Peto Left-censored k sample test). We detected lesions of relatively low severity in lung of challenged animals, whereas no lesions were observed in nasal turbinate or brain in any group (Fig. 8e, and Supplementary Fig. 5b). Accordingly, we only detected NP by immunohistochemistry in lung tissue on day 3 post-challenge, which was reduced in S-V987H vaccinated mice compared with challenged control animals, and on day 7 in the control group (Fig. 8f and Supplementary Fig. 5c) ($p < 0.05$ asymptotic generalized Pearson Chi-Squared test corrected for multiple comparison using FDR). Consistently, low levels of IVs were detected on day 3 in lung S-2P (3 out of 6), S-V987H (one out of 6) and control challenged mice (5 out of 6) (Supplementary Fig 5d). Thus, S-V987H-immunized K18-hACE2 mice challenged with the SARS-CoV-2 Omicron BQ1.1 variant showed an accelerated viral clearance.

## Discussion

SARS-CoV-2 uses the S glycoprotein to infect susceptible cells through a complex process that involves binding to the ACE2 receptor and subsequent structural reorganization. Hence, antibodies that target this protein and block its interaction with the ACE2 receptor or hamper its structural rearrangement could prevent infections[5]. The S is also a major target of both CD4+ and CD8 + T cell responses[25]. Therefore, most of the available SARS-CoV-2 vaccines use this protein as immunogen. However, the structural plasticity of the S may limit its capability to induce NAbs, similar to what has been shown for other functional equivalent proteins in other viruses (e.g., respiratory syncytial virus and human immunodeficiency virus 1)[11]. To date, several strategies have been used to stabilize the S glycoprotein in its prefusion conformation. Among them, the incorporation of two proline mutations between the HR1 and the central helix within the S2 subdomain stabilized the S glycoprotein of several coronaviruses, including MERS-CoV and SARS-CoV-2[12,13]. Such modifications result in increased production yields and improved immunogenicity[12,14]. In fact, some SARS-CoV-2 vaccines are mainly based on the 2P-stabilized S glycoproteins (i.e., mRNA–1273, BNT162b2 or Ad26.COV2.S)[3]. Unfortunately, the yields of the S-2P remains low under culture conditions (about 0.5 mg/L[13]), hampering its production and expression as a recombinant protein. To evaluate alternatives to the 2P strategy, we performed an in silico screening of a set of non-proline mutations based on their capability for maintaining a closed prefusion conformation of the Spike. In line with ref. 26, we observed that mutations that close the Spike glycoprotein had a detrimental impact on protein yield, suggesting that the open conformation may contribute to the expression of the protein. We also observed that the V987H mutation increased the production of the recombinant protein by two-fold and improved the ACE2 receptor recognition, suggesting a better RBD exposure. Next, we evaluated the immunogenicity of S-V987H, S-2P and recombinant RBD proteins in mice (C57BL/6 and K18-hACE2) and GSH, and their prophylactic capability in K18-hACE2 mice and GSH. While K18-hACE2 usually develop a severe form of viral-induced

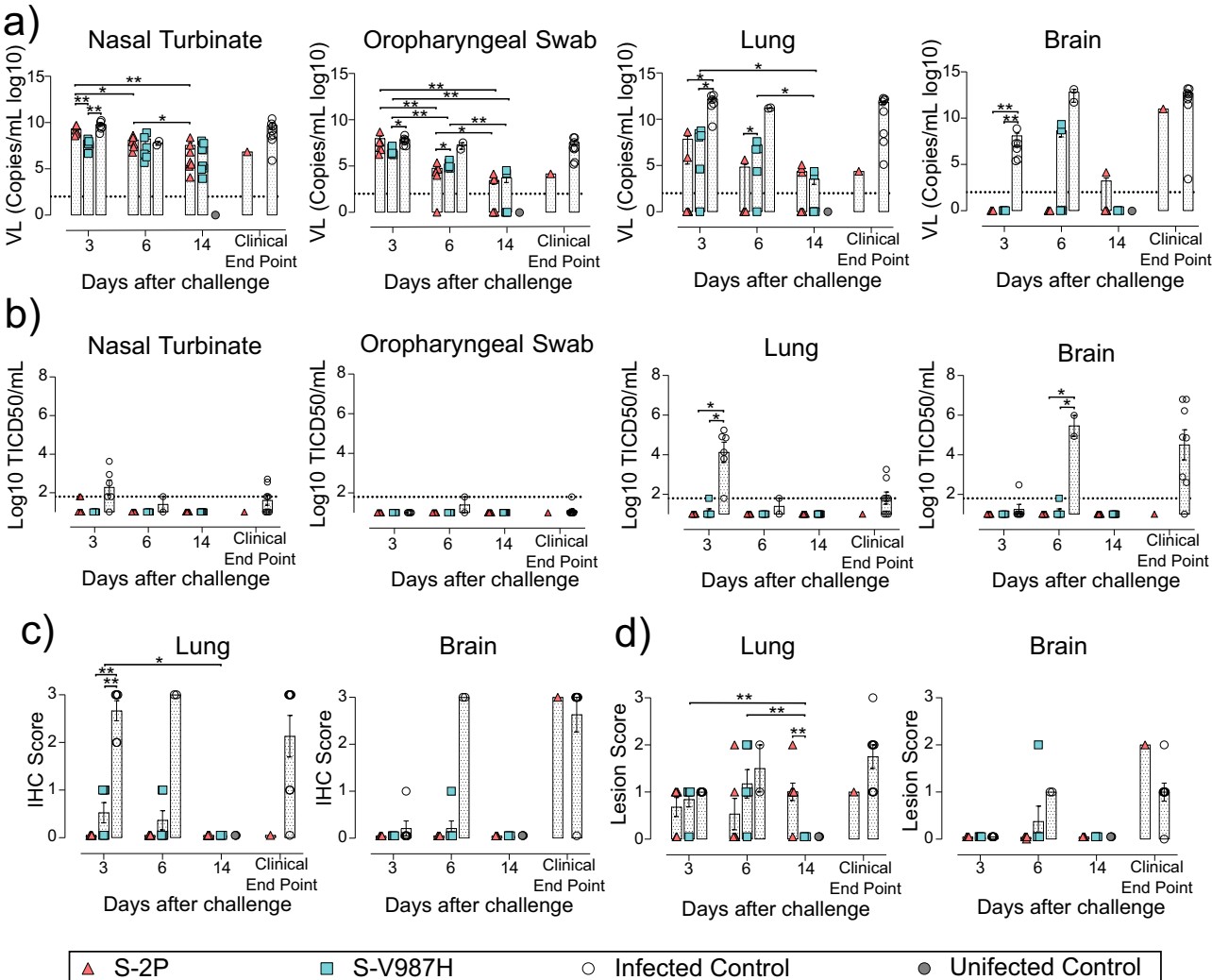

**Fig. 7 | Viral load and histopathology analysis of tissue samples from SARS-CoV-2 B.1.351 infected K18-hACE2 mice after vaccination.** SARS-CoV-2 viral load was analyzed in oropharyngeal swabs, and samples from nasal turbinate, lung and brain of infected K18-hACE2 mice. Virus distribution and tissue damage was analyzed by immunohistochemistry and histopathology. S-2P group: $n = 6$ on days 3 and 6; $n = 8$ on day 14, $n = 1$ clinical endpoints. S-V987H: $n = 6$ on days 3 and 6, n = 9 on day 14. Infected control: $n = 6$ on day 3, $n = 2$ on day 6, and $n = 8$ on clinical endpoints. Uninfected control: $n = 10$ on day 14. **a** Levels of SARS-CoV-2 gRNA (expressed as log10 of copies/mL) in oropharyngeal swabs, nasal turbinate, lung, and brain during infection. Dot line indicates limit of positivity (100 copies/mL). Differences between groups were analyzed using two-sided Peto & Peto left-censored k sample test with FDR correction. **b** Titer of infectious virions

determined in samples from nasal turbinate, oropharyngeal swab, lung and brain. Data are shown as Log10 of Median Tissue Culture Inhibition Dose per mL (TCID50/mL). Data were analyzed as indicated in (**a**). **c** Detection of SARS-CoV-2 nucleocapsid protein in brain, lung, and nasal turbinate by immunohistochemistry. Staining score: (0) no, (1) low, (2) moderate, and (3) high amount of viral antigen. **d** Histopathological analysis of brain, lung, and nasal turbinate by hematoxylin and eosin staining. Lesion score: (0) no, (1) mild, (2) moderate, and (3) severe lesion. Differences between groups were analyzed using two-sided Asymptotic Generalized Pearson Chi-Squared test with FDR correction. Mean plus standard error of the mean (SEM) is shown in (**a, b, c**, and **d**). * $p < 0.05$, **$p < 0.01$. Data represented in Fig. 7 correspond to one experiment. Source data are provided as a Source Data Fig. 7.

disease after challenge, and succumb to infection (excepting with most Omicron variants), GSHs progress to a moderate disease and spontaneously recover[17,27]. In addition to SARS-CoV-2 D614G variant, we also evaluated the protective efficacy of the S-V987H immunogen in K18-hACE2 mice challenged with the Beta and Omicron BQ1.1 variants. We chose the Beta VoC since it showed higher resistance against antibodies generated after vaccination or infection[21], and had increased virulence in K18-hACE2 mice[22]. BQ1.1 was selected as Omicron representative, since it has been shown as virulent in these mice[24]. Since adjuvants and the immunization strategy may impact on immunogenicity and efficacy of vaccines[28], we performed a heterologous prime/boost immunization strategy combining DNA (prime) and recombinant protein formulated with aluminum as adjuvant (boost), and two homologous administrations of the according recombinant protein formulated with AddaVax (MF59-like).

Even though S-V987H showed higher yields than S-2P, its immunogenicity was equivalent. Vaccination with the S trimers elicited higher humoral responses than the recombinant RBD protein, showing enhanced protective capability against severe disease. However, while all K18-hACE2 mice immunized with S-V987H were disease free, two mice in the S-2P group (one in the SARS-CoV-2 D614G challenge experiment and another one when SARS-CoV-2 Beta was used for challenge) developed severe disease. Thus, it is possible that S-V987H protective efficacy might be higher than the one observed with S-2P. Accordingly, mice immunized with S-V987H showed a faster viral clearance in respiratory tissues, which was confirmed in the SARS-CoV-2 Omicron BQ1.1 challenge experiment. In addition, we also evaluated the immunogenicity and prophylactic capability of these trimers in GSH, confirming the results obtained in the transgenic mouse model. Thus, our broad in vivo immunogenicity analysis of the S-V987H has

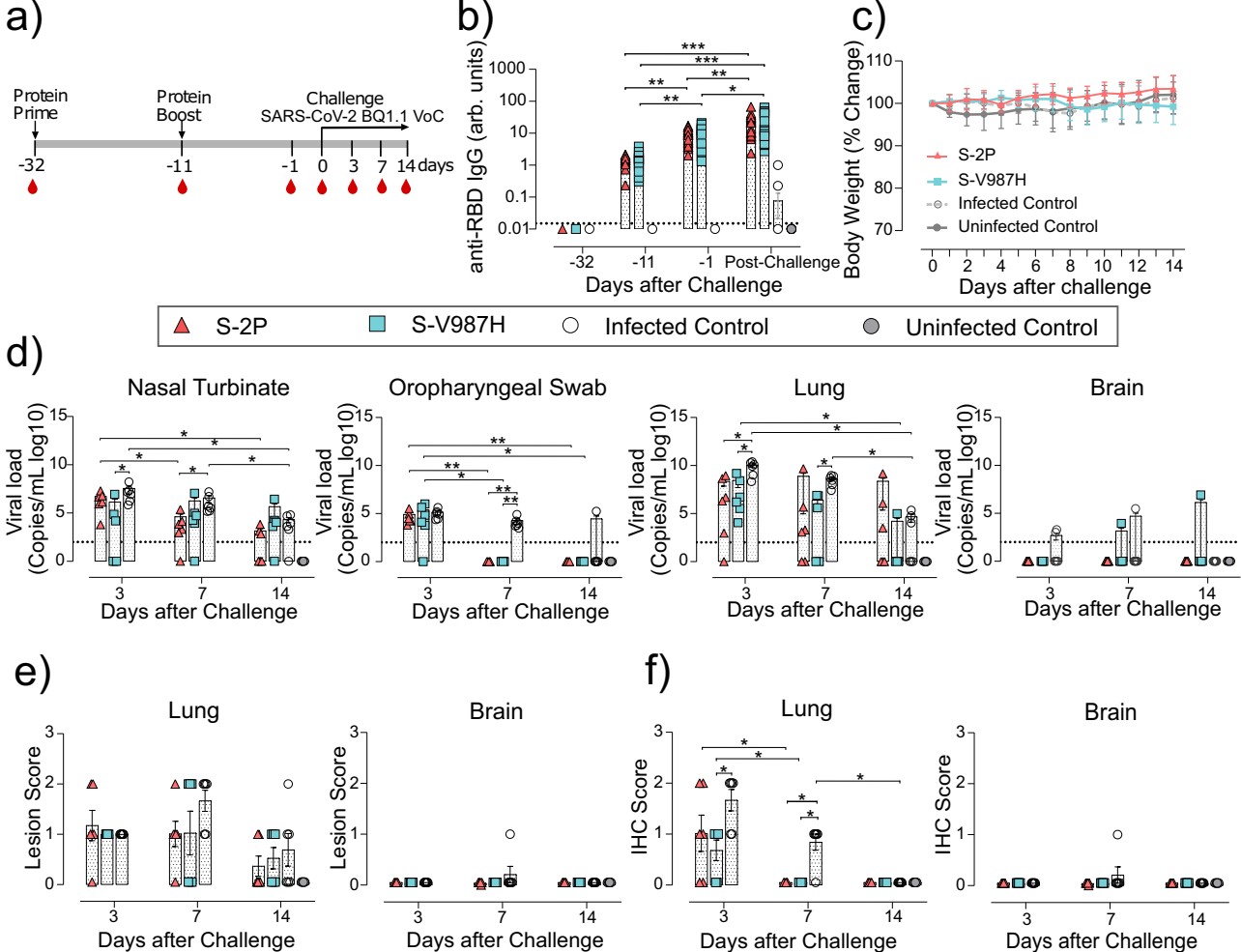

**Fig. 8 | Prophylactic activity of S-V987H immunization and analysis of the tissue viral load and humoral responses elicited in vaccinated K18-hACE2 mice challenged with the SARS-CoV-2 Omicron BQ1.1 variant.** K18-hACE2 mice were immunized twice with S-V987H or S-2P, adjuvanted with AddaVax. Then, mice were challenged with the SARS-CoV-2 Omicron BQ1.1 VoC. The humoral response, weight changes and survival of mice were evaluated after immunization and/or viral challenge. Tissue damage and viral loads were also analyzed. **a** Overview of vaccine strategy and infection timeline. Blood drops indicate collection of biological samples. **b** Kinetics of anti-RBD antibodies in serum samples expressed as arbitrary units (arb. units). Red triangles: S-2P group ($n = 18$). Blue squares: S-V987H group ($n = 18$). White circles: unvaccinated-challenged mice ($n = 18$) (Infected control). Gray circles: uninfected and unvaccinated mice ($n = 6$) (Uninfected control). Groups in each time point were analyzed using two-sided Conover-Iman test with multiple comparison correction by FDR. Differences among animals within a particular group along time were analyzed using two-sided Friedman test with FDR correction. **c** Percentage of weight variation in SARS-CoV-2 B.1.351 infected K18-hACE2 mice over time. Statistical analysis was performed against the unvaccinated

and challenged group using two-sided Kruskal-Wallis test with FDR correction. S-2P, S-V987H, and unvaccinated-challenged mice: $n = 18$ on days 0–3, $n = 12$ on days 4–6, $n = 6$ on days 7–14. Uninfected and unvaccinated mice: $n = 6$ on days 0–14. **d** Levels of SARS-CoV-2 gRNA (expressed as logarithmic of copies/mL) in oropharyngeal swabs, nasal turbinate, lung, and brain during infection. Dot line indicates limit of positivity (100 copies/mL). Differences between groups were analyzed using two-sided Peto & Peto left-censored k sample test with FDR correction. **e** Histopathological analysis of lung and brain by hematoxylin and eosin staining. Lesion score: (0) no, (1) mild, (2) moderate, and (3) severe lesion. **f** Detection of SARS-CoV-2 nucleocapsid protein in lung and brain by immunohistochemistry. Staining score: (0) no, (1) low, (2) moderate, and (3) high amount of viral antigen. Differences between groups in (**e**) and (**f**) were analyzed using two-sided Asymptotic Generalized Pearson Chi-Squared test with FDR correction. Statistically significant differences are indicated as follows: $*p < 0.05$, $**p < 0.01$, $***p < 0.001$. Mean plus standard errors of the means (SEM) are shown. Samples distribution in (**d**, **e** and **f**) is as described in (**b**). Data represented in Fig. 8 correspond to one experiment. Source data are provided as a Source Data Fig. 8.

shown that this mutation may be a good alternative to S-2P, since the incorporation of this mutation increased protein yields and improved protection in animal models.

Despite ref. 24. showed that SARS-CoV-2 Omicron BQ1.1 induced weight loss in K18-hACE2 mice upon infection with a viral dose of $10^4$ TCDI50, we did not observe this weight reduction, probably because our inoculation dose was lower ($10^3$ TCID50). We chose this viral dose for consistency with our previous challenge experiments with SARS-CoV-2 D614G and Beta variants.

The 2P strategy stabilized the S glycoprotein in its prefusion state by preventing conformational rearrangement[12]. However, this protein exists in a dynamic reversibly equilibrium between two different states:

close and open[29]. We hypothesized that the V987H mutation might alter this equilibrium favoring a trimer open state via steric effects, as we did not observe in our in silico models any stabilizing interaction caused by the incorporation of this mutation in the open conformation that could directly explain the preference of this variant for such state. In addition, V987H mutation might prevent large solvent exposition of the valine hydrophobic side chain in the open state, and reduce the putative instability generated by the two acidic residues flanking this position.

In addition to the 2P strategy, other approaches have been implemented to increase the stability and production of the S glycoprotein. In this sense, the inclusion of four-additional proline

mutations into the S2 subdomain of the S-2P protein (HexaPro Spike or S-6P) increased protein yields by 10-fold and improved its thermal stability[30]. In addition, Sun et al. showed that Newcastle disease virus expressing the S-6P protects GSH from SARS-CoV-2 induced disease[31]. Following a different approach, ref. 32 described a set of four mutations that stabilized the Spike protein in a closed conformation, increasing its expression by 6.4-fold. Remarkably, this protein did not require a heterologous trimerization domain. In addition, Riley and coworkers cross-linked the S2 subunit with different regions of the S1 subunit by introducing disulfide bridges, which also limited the motility of the RBD[33]. Besides these structure-guided mutation selection strategies, Tan et al. developed a high-throughput methods for the identification of pre-fusion stabilizing mutations within the heptad repeat 1 and central helix regions of the S2 subunit[34]. However, whether all these S protein variants show an improved immunogenicity or protective capabilities compared to the S-2P protein remains unknown. While Kalnin et al. observed that the immunogenicity of the S-2P immunogen was superior to the Hexa-Pro when they were assayed as mRNA vaccine[35], Lu and collaborators described a higher immunogenicity and efficacy of the S-6P protein compared with the S-2P when they were expressed in a VSV replicative vector[36]. Interestingly, in the latest study, the authors showed that S-6P was incorporated five times more efficiently on VSV particles than the S-2P protein. Thus, whether the observed enhancement in efficacy was due to an improved immunogenicity of S-6P vs S-2P or to a higher amount of S-6P protein in vaccine preparation would deserve further investigation.

Overall, here we described a S2-mutation (V987H) that improves the expression of the recombinant Spike glycoprotein. This protein was able to induce SARS-CoV-2 neutralizing antibodies and protect animals against both SARS-CoV-2 D614G and the highly pathogenic Beta variant. In addition, S-V987H immunization accelerate viral clearance after SARS-CoV-2 Omicron BQ1.1 challenge in K18-hACE2 mice. However, the present work shows some limitations: i) the improved expression of S-V987H seems to depend on the WH1 background since when it was evaluated in the Omicron BA4/5 context, this improvement was not observed (data not shown); ii) due to the high degree of clinical protection observed with the S-2P protein in the different studies, a formal demonstration of clinical superiority for any vaccine, determined as survival increase, would need an unfordable large amount of animals.

## Methods

### Recombinant trimeric Spike glycoprotein design and modeling

Unsolved secondary structures of the trimer in closed (PDB: 6VXX; https://www.rcsb.org/structure/6vxx) and open (PDB: 6VYB; https://www.rcsb.org/structure/6VYB) conformations[8] were reconstructed using SwissModel[37]. Then, all possible single mutations in both conformations were modeled using FoldX[38]. Comparison of the Gibbs free energy changes upon mutation between the open ($\Delta\Delta G_{open}$) and closed ($\Delta\Delta G_{closed}$) conformations ($\Delta\Delta G$) revealed a set of mutations predicted to strengthen the closed conformation. All single mutations predicted with $\Delta\Delta G < -1$ kcal/mol were rationally addressed by inspecting the three-dimensional models. In this regard, the final selection was based in two steps: i) selection of mutations predicted to increase the stability of the closed-conformation over the open-conformation using FoldX, ii) selection of mutations creating well-defined intermolecular interactions between the RBDs (including hydrophobic bonds, π−π interactions and cation-π interactions, ionic bonds, hydrophobic contacts or cavity filling mutations) that would exert a negative impact on the opening motion of the trimer.

### Recombinant protein production, quantification, purification, and ACE2 binding

The design of recombinant Spike glycoprotein is based on the one described by Wrapp[13]. Briefly, the C-terminal end of the extracellular portion of the S glycoprotein was fused to a T4-foldon trimerization domain followed by an 8xHis tag and a strep tag II. Plasmids for the expression of the recombinant S glycoproteins were obtained from GeneART (ThermoFisher scientific) in a pcDNA3.4 backbone. Proteins were produced by transient transfection using the ExpiFectamine 293 transfection kit (A14524, ThermoFisher Scientific) and following manufacturer specifications. Cell culture supernatants were harvested 5 days after transfection, clarified by centrifugation (3000 × g for 20 min) or using Sartoclear Dynamics® Lab V (Sartorius) and filtered at 0.2 μm using Nalgene Rapid-Flow sterile single use vacuum filter units (566-0020, ThermoFisher Scientific). Equivalent transfection efficiency was obtained for all the tested S variants. Proteins were purified by Immobilized Metal Affinity Chromatography using the HisTrap™ Excel columns (17-3712-06, Cytiva), and an ÄKTA start protein purification system (29022094, Cytiva). Purified proteins where concentrated and buffer exchanged to phosphate buffer saline by ultrafiltration (C7715, Merck Millipore) and stored at −80 °C until use. Integrity and purity of purified proteins were analyzed by sodium dodecyl sulfate polyacrylamide gel electrophoresis (SDS-PAGE) and Coomassie G-250 staining (NP0326BOX and LC6060, respectively, ThermoFisher Scientific) using a ChemiDoc MP Imaging System and the Image Lab Touch Software (12003154, Biorad).

Cell culture supernatants containing recombinant S-trimer variants, as well as purified proteins were quantified by ELISA. Briefly, Nunc MaxiSorp ELISA plates (M9410-1CS, Merck-Millipore) were coated overnight at 4 °C with 100 ng/well of HIS.H8 (MA1-21315; ThermoFisher Scientific), an anti-6xHis monoclonal antibody, in PBS. The following day, plates were blocked with PBS/1% of bovine serum albumin (MACS BSA, 130-091-376, Miltenyi Biotec) for 2 h at room temperature. Commercial His-Spike (40589- V08B1, Sino Biological) was used as standard, starting at 1 μg/ml and eight 1/3 serial dilutions. S variants were prepared in blocking buffer at 1/10 serial dilution for quantification. Samples were incubated overnight at 4 °C. After that, plates were washed and incubated for 2 h at room temperature with rabbit anti-SARS-CoV-2 Spike S2 IgG (40590-T62; SinoBiological) at 1/1000 dilution. Then, the HRP conjugated donkey anti-rabbit IgG (H + L) antibody (711-035−152; Jackson ImmunoResearch) at 1/10,000 dilution was used as detection antibody. Plates were revealed with o-Phenylenediamine dihydrochloride (P8787-100TAB, Sigma Aldrich) and stopped using 4N of $H_2SO_4$ (258105-1L-PC-M, Sigma- Aldrich). The signal was analyzed as the optical density (OD) at 492 nm with noise correction at 620 nm.

Binding of recombinant S-trimer mutated proteins to ACE2 was determined by ELISA. ELISA plates were coated with the HIS.H8 antibody and prepared as indicated above. After blocking, each sample was added in triplicate and incubated overnight (4 °C) at 0.1 μg/mL. After washing, a protein A-purified ACE2-human IgG Fc-fusion protein (homemade) was diluted in blocking buffer at 0.1 μg/mL, added to each well and incubated for two hours at room temperature. The HRP conjugated- (Fab)2 goat anti-human IgG (Fc specific) (115-036-071; Jackson ImmunoResearch) was used at 1/10,000 dilution as detection antibody and plates were revealed using OPD (P8787-100TAB, Sigma Aldrich). The enzymatic reaction was stopped adding 4 N of $H_2SO_4$ (258105-1L-PC-M Sigma Aldrich). Signal was analyzed as the OD at 492 nm with noise correction at 620 nm using an EnSight Multimode Plate Reader and the Kaleido Data Acquisition and Analysis Software (PerkinElmer). Binding to ACE2 was normalized according to protein concentration and represented as fold change related to S-2P.

### Cell culture, viral isolation, and titration

Vero E6 cells (ATCC CRL-1586) were cultured in Dulbecco's modified Eagle medium (DMEM-HPSTA, Capricorn Scientific) supplemented with 10% fetal bovine serum (FBS; A5256701, ThermoFisher Scientific), 100 U/mL penicillin, 100 μg/mL streptomycin (15140122, ThermoFisher Scientific).

The SARS-CoV-2 isolates used in the present work (Cat 01 and Cat 24) have previously been described[39,40] and were isolated, along with the Cat51 virus, from clinical nasopharyngeal swabs, as previously described in ref. 40. Viral isolates were subsequently grown in Vero E6 cells and sequenced as indicated below. Sequences were deposited at GISAID. Cat 01 is a SARS-CoV-2 D614G variant (ID EPI_ISL_510689), Cat 24 is a SAR-CoV-2 B.1.351 variant (originally detected in South Africa; EPI_ISL_1663571), and Cat51 is a SARS-CoV-2 Omicron BQ1.1 variant (ID EPI_ISL_16375266). Genomic sequencing was performed from viral supernatant by using standard ARTIC v3 or v4 based protocols followed by Illumina sequencing (https://doi.org/10.17504/protocols.io.bhjgj4jw). Raw data analysis was performed by viralrecon pipeline (https://github.com/nf-core/viralrecon) while consensus sequence was called using samtools/ivar at the 75% frequency threshold. Viral stocks and infectious viral particles in tissue samples from challenged animals (see below) were titrated in tenfold serial dilutions on Vero E6 cells to calculate the TCID50 per mL.

**In vivo immunization and challenge experiments.** All animal procedures were performed under the approval of the Committee on the Ethics of Animal Experimentation of the IGTP (Institute for Health Science Research Germans Trias i Pujol) and the authorization of Generalitat de Catalunya (Codes: 10965, 11221 and 11094). Prophylactic activity of the recombinant S-2P, S-V987H trimeric proteins and a recombinant monomeric RBD against SARS-CoV-2 D614G isolate (Cat01 isolate) was assessed in B6. Cg-Tg(K18-ACE2)2Prlmn/J (K18-hACE2) mice (stock #034860, Jackson Laboratories) and Golden Syrian hamsters (GHS) (8904, Envigo). Immunogenicity studies were also performed in C57BL/6JOlaHsd (5704, Envigo). In addition, protection against the SARS-CoV-2 B.1.351 (Beta) (Cat024 isolate) and SARS-CoV-2 Omicron BQ1.1 (Cat51 isolate) variants was evaluated in K18-hACE2 mice. The colony of these mice was maintained by breeding K18-hACE2 hemizygotes with C57BL/6J mice following the instructions of Jackson Laboratory (https://www.jax.org/strain/034860). Mice genotyping was performed according to the protocol 38170: Probe Assay - Tg(K18-ACE2)2Prlmn QPCR version 2.0 (https://www.jax.org/Protocol?stockNumber=034860&protocolID=38170). The GSH colony was maintained by brother/sister mating. Both mice and GSH colonies were stablished at the Centre for Comparative Medicine and Bioimage (CMCiB). Animals were maintained in cycles of 12 h light and 12 h dark, with controlled ambient temperature (22 °C) and relative humidity (30–70%), and with continue access to water and food. Environment enrichment elements, such as wood shavings, tissue paper and cardboard tubes for mice were used. Although the animal studies were not designed to investigate sex associated differences, we included balanced male and female groups when possible. No differences between sexes were observed. All data are represented and analyzed without sex distinction.

For the SARS-CoV-2 D614G challenge, 68 K18-hACE2 mice (50% male/50% female, 7–9 weeks old) were distributed in five experimental groups: S-2P ($n = 16$), V987H ($n = 14$), RBD ($n = 15$), infected positive controls ($n = 19$) and unvaccinated/uninfected negative control ($n = 4$). Sixty-eight GSH (male and female) (7–9 weeks old) (Envigo) were distributed in S-2P, S-V987H, RBD ($n = 16$/group, 8 males/8 females) and infected positive controls ($n = 16$/group, 15 females/1 male) and unvaccinated/uninfected negative control ($n = 4$, 2 females/2 males). Mice and GSH from the S-2P, S-V987H, and RBD groups were DNA-immunized by electroporation in the quadricep posterior. Forty microgram of plasmid coding for the corresponding immunogens were used (at 1 mg/mL). Animals were electroporated using a NEPA21 electroporator and tweezer electrodes (Nepagene). Two (for mice) or four (for GSH) weeks later, DNA-immunized animals received a boosting dose (40 μL) consisting of 15μg of recombinant protein adjuvanted with 20 μl of Adjust-Phos (vac-phos-250, Invivogen) in the

hock[41]. Control animals were primed with an empty vector and boosted with PBS+Adjust-Phos. Two weeks (mice) or 10 days (GSH) after boosting, animals were intranasally challenged with 1000 (mice) or 10,000 (GSH) TCID$_{50}$ of SARS-CoV-2 (Cat01 isolate) and followed up for 7 days. Weight and clinical signs were monitored daily after infection. Four animals for each experimental group, except for uninfected controls, were euthanized before challenge and on days 2, 4, or 7 post infection. Uninfected controls were euthanized on day 7 post-infection and only two mice from groups S-V987H and RBD were euthanized before challenge. However, any animal that showed a reduction of weight higher than 20%, a drastic reduction of mobility or a significant reduction of the response to stimuli were euthanized according to the humane endpoints defined in the supervision protocol. After euthanasia, oropharyngeal swab, nasal turbinate, lung, and brain were collected for viral load determination and histopathological analysis. Blood samples were collected before each immunization and viral challenge, and at euthanasia. Blood was left at room temperature for 2 h for clotting and serum was collected after centrifugation (10 min at $5000 \times g$) and stored at −80 °C until use.

For the SARS-CoV-2 B.1.351 (Beta) VoC challenge, 70 K18-hACE2 mice (50% male) were distributed as follows: S-2P ($n = 21$), S-V987H ($n = 21$), infected positive controls ($n = 16$) and unchallenged controls ($n = 10$). Mice from S-2P and S-V987H groups were immunized twice (40 μL/dose) (spaced 3 weeks between both doses) with 15 μg of recombinant protein adjuvanted with 20 μL of AddaVax (vac-adx−10, Invivogen) in the hock. Control mice received only PBS+AddaVax. Two weeks after the booster, mice from groups S-2P, S-V987H and challenged-controls were inoculated intranasally with 1000 TCID50 of SARS-CoV-2 Beta VoC (Cat24 isolate).

For the SARS-CoV-2 Omicron BQ1.1 VoC challenge, 60 K18-hACE2 mice (50% female) were distributed as follows: S-2P ($n = 18$), S-V987H ($n = 18$), infected positive controls ($n = 18$) and unchallenged controls ($n = 6$). Immunization and viral challenge (SARS-CoV-2 Omicron BQ1.1; Cat51) were performed as described for SARS-CoV-2 Beta variant.

**Quantification of anti-Spike antibodies by ELISA**
IgG antibodies elicited against the Spike and RBD glycoproteins were determined using and in-house ELISA in serum samples obtained from animals before each immunization and before viral challenge. In addition, humoral response was also evaluated in serum samples obtained from animals euthanized on days 2, 4, 7 after viral challenge or after humane endpoint. One half of a Nunc MaxiSorp ELISA plate was coated overnight at 4 °C with 50 ng/well of antigen in PBS [Spike (40589- V08B1) or RBD (40592-V08H), Sino Biologicals]. The other half-plate was incubated only with PBS. Then, the whole plate was blocked using PBS/1% of bovine serum albumin (MACS BSA, 130-091-376, Miltenyi Biotech) for two hours at room temperature. Mouse standards were prepared as seven 1/3 dilution of the anti-6xHis antibody HIS.H8 (MA1-21315; ThermoFisher Scientific), starting at 1 μg/mL. GSH standard was prepared similarly but using a positive GSH serum with the initial dilution at 1/100. All standards and samples were diluted in blocking buffer. After blocking, 50 μL of each standard or diluted samples were added to the antigen coated and antigen free wells in duplicate and incubated overnight at 4 °C. Each plate contained samples from all experimental groups. Plates were run in parallel to reduce inter-assay variability. After sample addition, plates were incubated overnight at 4 °C. The HRP conjugated (Fab)2 Goat anti-mouse IgG (Fc specific) (1/20,000) (115-036-071; Jackson Immunoresearch), or Goat anti-hamster IgG (H + L) (1/20,000) (107-035−142; Jackson ImmunoResearch) were used as detection antibodies for mouse and GSH IgG determination, respectively. Plates were revealed with o-Phenylenediamine dihydrochloride (P8787-100TAB, Sigma Aldrich) and stopped using 4 N of $H_2SO_4$ (258105-1L-PC-M, Sigma-Aldrich). The signal was analyzed as the OD at 492 nm with noise correction at

620 nm using an EnSight Multimode Plate Reader and the Kaleido Data Acquisition and Analysis Software (PerkinElmer).

The specific signal for each sample was calculated after subtracting the background signal obtained in antigen-free wells. Data is shown as arbitrary units (arb. units) according to the standard used.

## Neutralizing activity of serum samples

The neutralizing activity of serum samples was determined using HIV reporter pseudoviruses expressing SARS-CoV-2 S protein and Luciferase[42]. In brief, pseudoviruses were produced by co-transfecting Expi293F cells (A14527, ThermoFisher Scientific) with the pNL4-3.Luc.R-.E- (NIH AIDS Reagent Program[43]) and several SARS-CoV-2.SctΔ19 plasmids that code for the Spike glycoprotein of the WH1, Beta, Delta or Omicron variants. A VSV-G plasmid was used for the generation of VSV-G-pseudoviruses that were used as negative control. Transfections were performed using the ExpiFectamine293 Reagent kit (A14524, ThermoFisher Scientific). After 48 h, supernatants were harvested, filtered at 0.45 μm and frozen at −80 °C until use. Pseudoviruses were titrated on HEK293T cells overexpressing human ACE-2 (HEK293T/hACE2) (Integral Molecular).

Serum samples were inactivated at 56 °C for 60 min before use. Inactivated samples were 1/3 serially diluted in cell culture medium (DMEN, 10% fetal bovine serum) (range 1/100–1/24300) before mixing with 200 TCID50 of SARS-CoV-2 derived pseudoviruses and incubated for 1 h at 37 °C. Then, $2 \times 10^4$ HEK293T/hACE2 cells treated with DEAE-Dextran (Sigma-Aldrich) were added. After 48 h BriteLite Plus Luciferase reagent (6066766, PerkinElmer) was added and the results read in an EnSight Multimode Plate Reader. Data were calculated using a 4-parameters logistic equation in Prism 8.4.3 (GraphPad Software) and showed as normalized ID50 (reciprocal dilution inhibiting 50% of the infection). This assay has been previously validated with a replicative viral inhibition assay[44].

## Assessment of Spike-specific T cell response by interferon-γ ELISPOT

T cell responses against the SARS-CoV-2 Spike glycoprotein were determined by an IFN-γ-producing T cells ELISPOT assays (3321-2A; Mabtech) using splenocytes isolated from immunized C57BL/6 mice ($n = 10$/group, 50% female, 6–7 weeks old) (Envigo) 2 weeks after the second immunization. ELISPOT plates (Merck Rahway, NJ, USA) were coated with the purified anti-IFN-γ antibody (clone AN-18) at 2 μg/mL and incubated at 4 °C overnight. Splenocytes from immunized animals were seeded at $0.4 \times 10^6$ cells/well in RPMI-1640 media supplemented with 10% fetal bovine serum, penicillin (50 U/mL), and streptomycin (50 μg/mL) (15140122, ThermoFisher Scientific). The cells were stimulated overnight with a pool of S peptides (130-126-700, Miltenyi Biotech). Unstimulated cells were used to establish the basal IFN-γ T cell production. As a positive control, $0.2 \times 10^6$ cells were stimulated with concanavalin A (Con A) at 0.7 μg/mL (L7647, Merck Millipore). Cells were cultured for 20 hours at 37 °C + 5% $CO_2$. Plates were washed and incubated with a biotinylated anti-mouse IFN-γ detection antibody (clone R4-6A2-biotin) (1/2000 dilution) for one hour at room temperature. After that, streptavidin-ALP was added to plates and incubated for another hour at room temperature. Wells were developed using the BCIP/NBT-plus substrate solution (1706432, Biorad). Wells were imaged, and spots were enumerated using an ImmunoSpot reader (Cellular Technologies Limited). Data are shown as positive spot/100000 splenocytes.

## Viral load quantification in oropharyngeal swab and tissue samples

Oropharyngeal swabs and samples from nasal turbinate, lung and brain (only mice) were collected immediately after euthanasia in 1.5 mL Sarstedt tubes containing DMEM media (DMEM-HPSTA, Capricorn Scientific) supplemented with penicillin (100 U/mL) and streptomycin

(100 μg/mL). Tissue samples were homogenized twice at 25 Hz for 30 s using a TissueLyser II and a 3 mm Tungsten bead (69997, QIAGEN). After centrifugation for 2 min at 2000 × g, supernatants were collected and stored at −80 °C until use.

RNA was isolated using the Viral RNA/Pathogen Nucleic Acid Isolation kit and a KingFisher instrument (A42352, ThermoFisher Scientific), or an IndiMag pathogen kit (SP947457, Indical Bioscience) on a Biosprint 96 workstation (QIAGEN) following manufacturer's instructions.

PCR amplification in mice was based on the 2019-Novel Coronavirus Real-Time RT-PCR Diagnostic Panel guidelines and protocol developed by the American Center for Disease Control and Prevention. Briefly, a 20 μL PCR reaction was set up containing 5 μL of RNA, 1.5 μL of N2 primers and probe (2019-nCov CDC EUA Kit, 10006770, Integrated DNA Technologies) and 10 μl of GoTaq 1-Step RT-qPCR (A6020, Promega). Thermal cycling was performed at 50 °C for 15 min for reverse transcription, followed by 95 °C for 2 min and then 45 cycles of 95 °C for 10 s (s), 56 °C for 15 s and 72 °C for 30 s in the Applied Biosystems 7500 or QuantStudio5 Real-Time PCR instruments (ThermoFisher Scientific). For absolute quantification, a standard curve was built using 1/5 serial dilutions of a SARS-CoV2 plasmid (2019-nCoV_N_Positive Control, 200 copies/μL, 10006625, Integrated DNA Technologies) and run in parallel in all PCR determinations. The viral load of each sample was determined in triplicate and mean viral load (in copies/mL) was extrapolated from the standard curve and corrected by the corresponding dilution factor. Alternatively, results are shown as cycle threshold (Ct) or 2-ΔCt.

SARS-CoV-2 subgenomic RNA (sgRNA) was performed as previously described[45] with the following primers (Forward; 5'-CGATCTCTTGTAGATCTGTTCTC-3'; Reverse, 5'-ATATTGCAGCAGTACGCACACAA-3') and probe (5'- FAM-ACACTAGCCATCCT-TACTGCGCTTCG-TAMRA-3').

Mouse *gapdh* gene expression was measured in duplicate for each sample using TaqMan gene expression assay (4331182, ThermoFisher Scientific) as amplification control.

SARS-CoV-2 genomic RNA (gRNA) detection in GSH was performed based on RT-PCR described by ref. 46, which was adapted to the AgPath-ID One-Step RT-PCR kit (AM1005, Life Technologies). This RT-PCR targets a fragment of the envelope protein gene using the following primers (Forward: 5'-ACAGGTACGTTAATAGTTAATAGCGT-3'; Reverse: 5'-ATATTGCAGCAGTACGCACACA-3') and probe (5'-FAM-ACACTAGCCATCCTTA CTGCGCTTCG-TAMRA-3'). Thermal cycling was performed at 55 °C for 10 min for reverse transcription, followed by 95 °C for 3 min and then 45 cycles of 94 °C for 15 s, 58 °C for 30 s. SARS-CoV-2 subgenomic RNA detection in GHS was done as it is described in ref. 45. The primers and probes are the same as in gRNA determination except for forward primer: 5'-CGATCTCTTGTA-GATCTGTTCTC-3'. Thermal cycling for sgRNA was performed at 55 °C for 10 min for reverse transcription, followed by 95 °C for 3 min and the 45 cycles of 95 °C for 15 s, 56 °C for 30 s. Results are shown as Ct or 2-ΔCt.

## Pathology and immunohistochemistry

SARS-CoV-2 NP was detected by IHC using the rabbit monoclonal antibody 40143-R019 (Sino Biological) at 1:15,000 dilution. For immunolabelling visualization, the EnVision®+ System linked to horseradish peroxidase (HRP), (K4065, Agilent-Dako) and 3,3'-diaminobenzidine were used. The amount of viral antigen in tissues was semi-quantitatively scored as indicated in refs. 16,18. The following score was used: 0: No antigen detection, 1-low, 2-moderate and 3- high amount of antigen (Supplementary Fig. 6a). Nasal turbinate and lung from mice and GSH, and brain from mice were collected on days 0 (before viral challenge), 2, 4, 7, or at clinical endpoint after viral challenge, fixed by immersion in 10% buffered formalin and embedded into paraffin blocks. The histopathological analysis was performed on

slides stained with hematoxylin/eosin and examined by optical microscopy. A semi-quantitative scored based on the level of inflammation (0-No lesion; 1-Mild, 2-Moderate or 3-Severe lesion) was established (Supplementary Fig. 6b) based on previous classifications[16,18].

## Statistical analysis

ELISA binding data and neutralizing activity in each time point were analyzed using two-tailed Connover Test with multiple comparison correction by false discovery rate (FDR). Differences among animals within a particular group along time were analyzed using the two-tailed Friendman test corrected for multiple comparison using FDR. Weight variation in SARS-CoV-2 challenged mice over time was analyzed using two-tailed Kruskal-Wallis test corrected by FDR. Severe disease incidence was represented by Kaplan-Meier plots. Statistical analysis was performed against unvaccinated group using two-tailed Mantel-Cox test. Levels of SARS-CoV-2 gRNA, sgRNA, and IV particles in tissues were analyzed using two-tailed Peto & Peto left-censored k sample test corrected by FDR. Histopathology analysis was carried out using two-tailed Asymptotic Generalized Pearson Chi-Squared test with FDR correction. $P$ values are indicated as follows: $*p < 0.05$, $**p < 0.01$, $***p < 0.001$, $****p < 0.0001$. Whereas $p$ values close to statistical significance are shown as number. Statistical analyses were conducted using the R (version 3.6) software environment and GraphPad Prism v8.0.

## Reporting summary

Further information on research design is available in the Nature Portfolio Reporting Summary linked to this article.

## Data availability

All data are included in the published version of the article, and its supplementary information files. Source data are provided with this paper. The PDB accession codes for the previously generated Spike trimer in close and open conformations are 6VXX and 6VYB, respectively. The previously generated sequences of the SARS-CoV-2 isolates used in the present work are available in GISAID with the following codes: Cat01 (SARS-CoV-2 D614G variant): ID EPI_ISL_510689; Cat24 (SAR-CoV-2 B.1.351): ID EPI_ISL_1663571; and Cat51 (SARS-CoV-2 Omicron BQ1.1): ID EPI_ISL_16375266. Source data are provided with this paper.

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

## Acknowledgements

This work was supported by Grifols pharmaceutical, the CERCA Program (2021 SGR 00452 to B.C.; Generalitat de Catalunya), Direcció General de Recerca i Innovació en Salut (Generalitat de Catalunya) (projects SLD0015 to J.C. and SLD0016 to J.B.), the Carlos III Health Institute (PI17/01518 to J.B. and PI18/01332 to J.C.). J.B. is supported by the Health Department of the Catalan Government (Generalitat de Catalunya). In addition, the project was also supported by the crowdfunding projects "YomeCorono" (to A.V., V.G., J.C., B.C., J.B. and N.I.-U.), BonPreu/Esclat, and Correos (to J.B.). CAN was supported by a predoctoral grant from Generalitat de Catalunya and Fons Social Europeu (2020 FI_B_0742). A.P.G. was supported by a predoctoral grant from Generalitat de Catalunya and Fons Social Europeu (2022 FI_B_00698). P.A.R. was funded by a predoctoral fellowship from the Government of Catalonia (2020FI_B2_00138). E.P. was supported by a doctoral grant from National Agency for Research and Development of Chile (ANID: 72180406). N.I.-U. is supported by the Spanish Ministry of Science and Innovation (grant PID2020-117145RB-I00), EU HORIZON-HLTH-2021-CORONA-01 (grant 101046118). This study was also supported by CIBER—Consorcio Centro de Investigación Biomédica en Red (CB 2021), Carlos III Health Institute, Ministerio de Ciencia e Innovació and Unión Europea—NextGenerationEU. We would like to thank Foundation Dormeur that support the acquisition of the QuantStudio-5 real time PCR system, an Eclipse Ts2R-FL Inverted Research Microscope, and an ÄKTA go protein purification system. Funders had no role in study design, data analysis, decision to publish, or manuscript preparation. We thank to Ismael Valera, the CMCiB's staff (Sara Capdevila, Jordi Grifols, Rosa Maria Ampudia, Jorge Diaz, Yaiza Rosales and Sergi Sunye) and the BSL3 IRTA-CReSA staff (Xavier Abad, Ivan Cordon, Anna Pou, Oscar García, Joanna Wiacek, Maria Angeles Osuna, Luís Ribas and Claudia Pereira Sunyé) for their technical assistance with in vivo animal studies.

## Author contributions

J.C., B.C., J.S., J.B., J.V.A., N.I.U., V.G. and A.V.: Study conception, design, and funding J.C., N.P.L., C.A.N., P.A.R.: Manuscript draft preparation C.A.N., B.T., P.A.R., E.A.E., M.B., N.P.L., M.L.R., V.U., JR., N.R., M.P. E.P., S.M., E.B., E.R.M., G.C., F.T.F., A.P.G., C.R., C.A., R.O., A.B., R.L., D.P.Z., J.M.B., N.I.U., J.B., J.V.A., V.G., J.S., and J.C.: Data acquisition, analysis, and interpretation.

## Competing interests

The present work counted on funding from Grifols pharmaceutical. However, Grifols had no role in study design, data analysis, decision to publish or manuscript preparation. P.A.R. is currently employed by Sanofi, which has no association with any content related to this work. The authors declare no other competing interests. Unrelated to the submitted work, J.B. and J.C. are founders and shareholders of AlbaJuna Therapeutics, S.L. B.C. is founder and shareholder of AlbaJuna Therapeutics, S.L. and AELIX Therapeutics, S.L, and V.G. is founder and shareholder of Nostrum Biodiscovery. Unrelated to the submitted work, N.I.-U. is supported by institutional funding from Pharma Mar, HIPRA, Amassence, and Palobiofarma.

## Additional information

[1]IrsiCaixa AIDS Research Institute, Campus Can Ruti, Badalona, Spain. [2]Unitat Mixta d'Investigació IRTA-UAB en Sanitat Animal, Centre de Recerca en Sanitat Animal (CReSA), Campus de la Universitat Autònoma de Barcelona (UAB), 08193 Bellaterra Barcelona, Catalonia, Spain. [3]IRTA Programa de Sanitat Animal, Centre de Recerca en Sanitat Animal (CReSA), Campus de la Universitat Autònoma de Barcelona (UAB), 08193 Bellaterra Barcelona, Catalonia, Spain. [4]Life Sciences Department, Barcelona Supercomputing Center (BSC), Barcelona, Spain. [5]Germans Trias i Pujol Research Institute (IGTP), Campus Can Ruit, Badalona, Spain. [6]CIBERINFEC. ISCIII, Madrid, Spain. [7]Centre for Health and Social Care Research (CESS), Faculty of Medicine. University of Vic—Central University of Catalonia (UVic—UCC), Vic, Catalonia, Spain. [8]Catalan Institution for Research and Advanced Studies, Barcelona, Spain. [9]Fundaciò Lluita contra les infeccions. Hospital Germans Trias i Pujol, Badalona, Catalonia, Spain. [10]Universitat Autonoma de Barcelona. Bellaterra, Cerdanyola del Vallès, Catalonia, Spain. [11]Departament de Sanitat i Anatomia Animals, Facultat de Veterinària, UAB, Bellaterra, Cerdanyola del Vallès, Spain. [12]Present address: Department of Biomedical Sciences, Institute of Tropical Medicine, Antwerp, Belgium. [13]These authors contributed equally: Carlos Ávila-Nieto, Júlia Vergara-Alert, Pep Amengual-Rigo. ✉e-mail: joaquim.segales@irta.cat; jcarrillo@irsicaixa.es

