## [Peer Review File · Nature Communications]

Immunization with V987H-stabilized Spike glycoprotein protects K18-hACE2 mice and golden Syrian hamsters upon SARS-CoV-2 infection.REVIEWER COMMENTS

Reviewer #1 (Remarks to the Author):

In this manuscript, Avila-Nieto and colleagues attempted to develop a structurally more stable SARS-CoV-2 spike protein by introducing a novel V987H mutation. The authors suggests the introduction of this mutation to the spike protein could limit the mobility of the protein structure and allow exposure of the RBD, and thus potentially enhance immunogenicity. They demonstrated the possibility of introducing this spike mutant as vaccine to better protect the laboratory animals against SARS-CoV-2 infection. Below are some comments that the authors may consider:

1. Is there any explanation on how the mutation V987H can maintain the "close" state and stabilize the RBD domain, provided that S-V987H and S-2P (K986P / V987P) only differ by two amino acids? Evidence on how S-V987H can maintain a more stable and exposed RBD than S-2P would be value-adding to this manuscript.
2. In the main text, it was mentioned 11 single mutations were selected for characterization (line 124). However, in Figure 1c and 1d, data of only 10 single mutations were shown. G416R was missing.
3. Vaccination regimen has great effect on efficacy. Is there any justification for the difference in vaccination regimens between experiments of SARS-CoV-2 D614G and Beta VOC viral challenge? Prior to D614G challenge, the animals were primed by DNA electroporation and boosted by recombinant protein, while two recombinant protein vaccine doses were given to the animal before Beta VOC challenge.
4. What is the route of administration for the booster dose of recombinant protein?
5. How do you account for the enhanced protection and viral clearance in mice immunized with S-V987H, provided that the humoral response elicited by S-2P and S-V987H were similar?
6. Please correct the typo. It should be "AddaVax" instead of "AddVax" (line 310).
7. What are the doses of the adjuvants used? Seems they have not been mentioned throughout the manuscript.
8. According to the results presented in Fig 6b, 6d (anti-RBD and nAb against Beta VOC) and Fig 7 (viral load in oropharyngeal swab, lungs and brain, and histopathology), it seems that the conclusion on S-V987H conferring better protection than S-2P group against Beta VOC due to higher serum neutralization is not well supported (line 360-363).
9. Other than humoral response, would S-V987H better elicit other immune responses that may improve the disease outcome compared to S-2P? T cell response may be worth characterization.

Reviewer #2 (Remarks to the Author):

In the manuscript, authors have identified a novel mutation in the SARS-CoV-2 spike protein to stabilize the spike for vaccine efforts. The authors identified mutation when used as a vaccine in mice and hamsters provided largely equivalent protection as the S2P. Overall the biggest improvement of their mutation seems to be protein production rather than vaccine efficacy.

Major comments:

Efficacy data against relevant VoCs is lacking. Although the disease may be milder in animal models (Line 399-400), authors could have more thoroughly evaluated neutralizing activity against contemporary VoCs. What is the neutralizing activity of vaccinated animals against omicron prior to challenge? Authors only show neutralizing activity post challenge. Disease is also not so mild as to preclude vaccine efficacy studies.

Did authors evaluate cellular immunity?

How well would the vaccine work in animals or people with pre-existing immunity? Virtually everyone has some level of existing immunity to the virus, either through vaccination, infection or both.

Efficacy in the upper respiratory tract seems minimal in the hamster model or with the Beta VoC. Do authors think the vaccine would prevent transmission?

Minor comments:

Why do authors show copies/mL for mouse data and then Ct values for hamster data?

Were the people scoring the lesions and presence of antigen blinded to study groups?

Reviewer #3 (Remarks to the Author):

Currently, most COVID-19 vaccines use a mutant variant of the spike glycoprotein known as S-2P, which has better stability, yield, and immunogenicity. However, production of S-2P remains limited. In this manuscript, the authors introduce a novel mutation, V987H, that significantly increases the production of recombinant spike protein and potentially improves the exposure of the receptor binding domain (RBD). The immunogenicity of the newly developed S-V987H variant has been shown to be comparable to S-2P and better than a monomeric RBD in both K18-hACE2 mice and Golden Syrian hamsters. Notably, immunization with S-V987H provides complete protection against severe disease, including resistance to the D614G and B.1.351 variants, in both animal models following SARS-CoV-2 infection. In addition, mice immunized with S-V987H show accelerated viral clearance in tissues compared with mice immunized with RBD or S-2P. The authors suggest that the S-V987H protein represents a promising alternative to S-2P for future SARS-CoV-2 vaccine development. Overall, the work takes a straightforward approach with a clear objective, and the results are concisely presented following standard experimental practices. However, I have several concerns that should be addressed in revising the manuscript.

COMMENTS

1. Although the spike protein used in this study was intentionally derived from the Wuhan-like strain of SARS-CoV-2, it would be valuable to demonstrate the applicability of the V987H mutation to Omicron variants, particularly the BA.5 spike included in the bivalent vaccine. Currently, spike proteins derived from the Omicron variant are known to have a lower yield compared to the original strain. Therefore, it would increase the importance of this manuscript if the potential of the V987H mutation to increase the yield of the Omicron spike could be explored and presented.
2. The S-2P variant was improved by the Hexa-Pro approach (DOI: 10.1126/science.abd0826), which involved the introduction of six prolines instead of two. It is conceivable that Hexa-Pro could replace S-2P in future spike protein designs. The authors mentioned briefly in the discussion but it is still not clear. To further strengthen the manuscript, the authors might consider addressing this point by comparing the results of their study with the yields reported in the literature based on the Hexa-Pro strategy.
3. Please provide more details in the manuscript about the experiment shown in Fig. 1D. The Material and Methods section does not provide sufficient information about this experiment. In line 133, the authors mention that "most" of the variants are in a closed trimer conformation. Please indicate which variants in the graph are considered to be in this particular conformation.
4. It would be useful to provide an explanation, or at least a comment, as to why replacing histidine with proline at amino acid position 987 might improve protein expression. In addition, it would be interesting to investigate whether the presence of K986P could affect the protein expression and/or immunogenicity of V987H.

5. Please elaborate in their manuscript on the rationale for using DNA prime as opposed to administering two doses of recombinant proteins. It is imperative to clarify whether the inclusion of DNA prime could introduce confounding factors when comparing the immunogenicity of the two spike constructs.

6. Please consider including measurements of infection titers such as TCID₅₀ or PFU in addition to the current methodology, which relies solely on viral genome copies to determine viral load. This data would allow for a more comprehensive assessment of viral replication dynamics.

MINOR COMMENTS

1. Please edit the title to avoid confusion that K18-hACE2 are mice, not hamsters.

2. Coomassie blue staining of the recombinant S constructs in Fig. 2A indicates higher expression of S-2P than of S-V987H. Please clarify whether the same amount of protein was loaded.

3. Please confirm that the unit of anti-RBD IgG (in AU/ml) in Fig.2C ranges from 0.1 to 10. This seems to be very low. Is it possible to convert it to standard units to compare with other studies?

REVIEWER COMMENTS

Reviewer #1 (Remarks to the Author):

In this manuscript, Avila-Nieto and colleagues attempted to develop a structurally more stable SARS-CoV-2 spike protein by introducing a novel V987H mutation. The authors suggests the introduction of this mutation to the spike protein could limit the mobility of the protein structure and allow exposure of the RBD, and thus potentially enhance immunogenicity. They demonstrated the possibility of introducing this spike mutant as vaccine to better protect the laboratory animals against SARS-CoV-2 infection. Below are some comments that the authors may consider:

1. Is there any explanation on how the mutation V987H can maintain the “close” state and stabilize the RBD domain, provided that S-V987H and S-2P (K986P / V987P) only differ by two amino acids? Evidence on how S-V987H can maintain a more stable and exposed RBD than S-2P would be value-adding to this manuscript.

It has been described that the SARS-CoV-2 Spike glycoprotein can adopt several prefusion conformations, modulating the RBD exposure. A dynamic conversion between close and open prefusion states have even been described in prefusion stabilized Spike glycoproteins, such as S-2P and S-6P. Thus, Costello et al. described that the Spike glycoprotein in a prefusion closed state (state A) can spontaneously evolve to an open prefusion conformation (state B) and vice versa (10.1038/s41594-022-00735-5). This transition can be modulated by temperature or even by little sequence changes and may have an important impact on the antigenicity of the spike, since cryptic neutralizing epitopes within the S2 trimer interface are better exposed in the state B (open trimer). As seen in Figure 1D, our V987H mutant promotes the open state, and might favor the Spike trimer in a state B conformation (open). Looking at the differences between the opened and closed crystals (see images below) we see several factors pointing to the stabilization of the open state by introducing a histidine in position V897: i) steric impediment in the closed state (notice the short distance between positions 987 and 413 from the next chain), ii) large solvent exposition of a hydrophobic side chain in the open state if valine occupies this position, and iii) two acidic residues flanking position 987 that largely increase the instability of a hydrophobic side chain (less important in a closed/shielded state).

Thus, although V987H is an exception in FoldX predictive pattern, it may stabilize the Spike in a more open conformation. While it is not possible to make accurate

predictions on how a single mutation might increase expression (best models such as NetSolP have been trained with full sequences), there have been described several evidence that correlate Spike stabilization with an increase in expression (i.e S-2P and S-6P proteins). We have included this explanation in the discussion section lines 501-509.

2. In the main text, it was mentioned 11 single mutations were selected for characterization (line 124). However, in Figure 1c and 1d, data of only 10 single mutations were shown. G416R was missing.

We would like to thank the reviewer for noting that. Figure 1c and 1d have been updated including the G416R mutation.

3. Vaccination regimen has great effect on efficacy. Is there any justification for the difference in vaccination regimens between experiments of SARS-CoV-2 D614G and Beta VOC viral challenge? Prior to D614G challenge, the animals were primed by DNA electroporation and boosted by recombinant protein, while two recombinant protein vaccine doses were given to the animal before Beta VOC challenge.

We totally agree with the reviewer's comment. In fact, since immunization strategy may impact immunogenicity and efficacy, we wanted to evaluate two different approaches: DNA+protein vs protein+protein, and two adjuvants widely used in human vaccines: aluminum, a Th2-prone adjuvant, and AddaVax, a MF59-like adjuvant that induces both Th1 and Th2 responses. Initially, we chose DNA + protein because it is well known that these immunization regimens induce both T and B cells responses. Once we confirmed that this strategy worked, we decided to test another system based on the AddaVax adjuvant, which also induce both T and B cells responses and whose administration would be easier than DNA immunization. In addition. Elderly people are a high risk of developing a severe COVID-19 upon SARS-CoV-2 infection. Since MF59 is used in vaccines administered to elder population (i.e., flu vaccine(10.1093/cid/ciac302)), we considered of interest to test an equivalent adjuvant in our experimental setting. Please, see also our reply to reviewer 3.

4. What is the route of administration for the booster dose of recombinant protein? Proteins were administered subcutaneously in the hock. This is a humane alternative to the footpad (10.1016/j.jim.2007.08.004).

5. How do you account for the enhanced protection and viral clearance in mice immunized with S-V987H, provided that the humoral response elicited by S-2P and S-V987H were similar?

Unfortunately, we do not have a clear explanation for that. Although binding assays showed that both S-2P and V987H induced similar levels of binding antibodies, it is still possible that V987H induced a better immune response against cryptic epitopes that are not detected in the binding assays. V987H slightly improved RBD exposure, which might contribute to the generation of distinctive antibodies. Another possibility that cannot be ruled out is that V987H-induced antibodies are more efficient mediated Fc-related function such as antibody-dependent cellular phagocytosis (ADCP) or cytotoxicity (ADCC).

6. Please correct the typo. It should be “AddaVax” instead of “AddVax” (line 310).
The typo has been corrected in the new version of the manuscript.

7. What are the doses of the adjuvants used? Seems they have not been mentioned throughout the manuscript.

We follow the supplier indications. We mix proteins and adjuvant in a 1:1 proportion. In each immunization we injected 40 μ L of protein-adjuvant mixes in the hock, therefore we used 20 μ L of adjuvant in each immunization. This information has been added to the methods section lines: 647-648, and 668-669.

8. According to the results presented in Fig 6b, 6d (anti-RBD and nAb against Beta VOC) and Fig 7 (viral load in oropharyngeal swab, lungs and brain, and histopathology), it seems that the conclusion on S-V987H conferring better protection than S-2P group against Beta VOC due to higher serum neutralization is not well supported (line 360-363).

The immunization route that we chose (subcutaneous vaccination) did not generate a sterilizing immunity but can prevent the development of severe disease. Therefore, after challenge, mice get infected, but the immune response controlled the viral replication protecting against disease development. That would explain why we are detecting viral gDNA in almost all analyzed tissues. However, it does not mean they are infectious virions. One feature of the K18-hACE2 mouse model is that upon SARS-CoV-2 challenge (with the infectious dose of the inoculum used here), mice develop a strong affectation of the central nervous system, which ultimately leads to death. Therefore, it is crucial that the vaccine induced immune response prevent viral spreading to central nervous system. Besides neutralization others immune mediated factors can also have an important role in protection. Among them, Fc-mediated effector functions, such as ADCC and ADCP, or T cell responses. Thus, we have modulated our statement in lines 402-404 as follows:

“To summarize, the immunogenicity of both S-2P and S-V987H trimers was similar in K18-hACE2 SARS-CoV-2 Beta-infected mice, although S-V987H promoted the development of higher serum neutralization. These antibodies along with other immune mediated effector functions, such as antibody dependent cellular phagocytosis or cytotoxicity, and T cell responses, might contribute to the increase in protection observed in S-V987H vaccinated animals, compared to the S-2P group.”

9. Other than humoral response, would S-V987H better elicit other immune responses that may improve the disease outcome compared to S-2P? T cell response may be worth characterization.

We agree with the reviewer that other immune functions, particularly T cell responses, can also contribute to protection. To investigate that point, we have performed two immunization studies in C57BL/6 mice. Please, see our reply to reviewer 2 for more information about T cell responses.

Reviewer #2 (Remarks to the Author):

In the manuscript, authors have identified a novel mutation in the SARS-CoV-2 spike protein to stabilize the spike for vaccine efforts. The authors identified mutation when used as a vaccine in mice and hamsters provided largely equivalent protection as the S2P. Overall the biggest improvement of their mutation seems to be protein production rather than vaccine efficacy.

To our knowledge this work is one of the few that analyze vaccine efficacy by a head-to-head comparison with the S-2P protein in animal models. S-2P showed improved yield and efficacy when it was compared with the wild type S protein. Vaccines based on the S-2P stabilization strategies such as RNA vaccines (Moderna and Pfizer), adenoviral vectors (Janssen) or recombinant protein (Sanofi) have demonstrated an impressive efficacy (higher than 90% of protection against severe disease in the case of mRNA vaccine). Therefore, the formal demonstration that a novel S stabilization strategy shows better efficacy than the S-2P one is technically challenging and would need of a very high number of animals. Therefore, we consider that the slight differences in efficacy showed in this work are encouraging and highly relevant.

Major comments:

Efficacy data against relevant VoCs is lacking. Although the disease may be milder in animal models (Line 399-400), authors could have more thoroughly evaluated neutralizing activity against contemporary VoCs. What is the neutralizing activity of vaccinated animals against omicron prior to challenge? Authors only show neutralizing activity post challenge. Disease is also not so mild as to preclude vaccine efficacy studies.

Since the blood collected from alive animals is low, we could not evaluate the neutralizing activity of serum samples before euthanasia. To overcome this experimental limitation, we have performed two additional immunization experiments in C57BL/6 mice repeating the immunization conditions described in Fig. 2b and Fig. 6a. Two weeks after the booster dose, mice were euthanized, and blood and spleen collected. We determined the neutralizing activity of these samples against SARS-CoV-2 BA4/5 and BQ1.1. Results are indicated in both Supplementary Fig 1f-g, and Supplementary Fig 4c, and in lines 185-194 and 359-365. They showed that both S-2P and S-V987H induced SARS-CoV-2 Omicron BQ1.1 and BA4/5 neutralizing antibodies, particularly when AddaVax was used as adjuvant, albeit at low levels.

In addition, we have performed an efficacy study in immunized K18-hACE2 mice challenged with the SARS-CoV-2 Omicron BQ1.1 variant. We have chosen this variant because Case et al. have demonstrated that it is pathogenic in K18-hACE2 mice (10.1128/jvi.00628-23), observing a weight reduction as for day 4 post challenge. In our experiment, we did not observe this weight lost, probably because we used a lower challenge dose (10^3 vs 10^4 TCID₅₀ used by case et al.). We used 10^3 TCID₅₀ for consistency with our previous experiment using SARS-CoV-2 D614G and Beta variants. Although we were not able to establish clinical differences among groups, our results support a faster viral clearance in S-V987H immunized mice. These results are showed

in lines 406-440 and in Fig 8 and Supplementary Fig 5.

Did authors evaluate cellular immunity?

We didn't analyzed T cell responses previously because our BSL3+ facility is not equipped with the instrumental needed for it. To determine the capability of our immunogens to elicited T cells responses, we have performed two additional immunization experiments in C57BL/6 mice following the same experimental conditions described in the manuscript (Figure 2b and 6a). In the first experiment, 3 groups of 10 mice (5 females/5 males) were immunized following a prime (DNA) boost (protein + aluminum hydroxide gel) with the immunogen S-2P, S-V987H or RBD, respectively. A group of 5 mice (2 female and 3 males) were used as negative controls. In the second experiment, two groups of 10 mice (5 females/5 males) were immunized twice with protein (S-2P or S-V987H) + AddaVax three weeks apart. A group of 5 mice (3 female and 2 males) were used as negative controls. T cell responses were analyzed by ELISpot, quantifying the number of SARS-CoV-2 IFN- γ -producing splenocytes. The results of the first experiment showed that S-2P and S-V987H elicited a higher T cell response than the RBD, but no differences were observed between S-2P and S-V987H groups. In the second experiment we confirmed the lack of differences between both trimers' immunized groups. However, a tendency in favor of a higher T cell response was observed in the S-V987H immunized group. No differences were observed between males and females. This new information has been added in the material and methods section (lines 728-745), and in in lines 195-198 (Supplementary Fib- 1h) 365-368 (Supplementary Fig. 4d).

How well would the vaccine work in animals or people with pre-existing immunity? Virtually everyone has some level of existing immunity to the virus, either through vaccination, infection, or both.

This is a relevant question. However, we have not conducted any experiment in this direction because it is out of the scope of the present work. We would expect that S-V987H can boost pre-existing immune responses, but this point needs to be empirically confirmed.

Efficacy in the upper respiratory tract seems minimal in the hamster model or with the Beta VoC. Do authors think the vaccine would prevent transmission?

Since we immunized mice subcutaneously, it is expected that we are not generating a sterilizing immunity at the viral entry site. Therefore, mice are infected and can transmit the virus, at least during the first days. However, it is also expected that since vaccinated animals control viral replication faster, the period of time in which they can still transmit the virus is shorter than in unvaccinated animals.

Minor comments:

Why do authors show copies/mL for mouse data and then Ct values for hamster data? This was because both determinations were performed in different labs (IrsiCaixa and IRTA-CRESA). We have now corrected them and gRNA data are shown as copies/mL (Figure 5a).

Were the people scoring the lesions and presence of antigen blinded to study groups?
Yes, all analyses were performed in a blinded fashion by a European College of Veterinary Pathologists (ECVP) specialist.

Reviewer #3 (Remarks to the Author):

Currently, most COVID-19 vaccines use a mutant variant of the spike glycoprotein known as S-2P, which has better stability, yield, and immunogenicity. However, production of S-2P remains limited. In this manuscript, the authors introduce a novel mutation, V987H, that significantly increases the production of recombinant spike protein and potentially improves the exposure of the receptor binding domain (RBD). The immunogenicity of the newly developed S-V987H variant has been shown to be comparable to S-2P and better than a monomeric RBD in both K18-hACE2 mice and Golden Syrian hamsters. Notably, immunization with S-V987H provides complete protection against severe disease, including resistance to the D614G and B.1.351 variants, in both animal models following SARS-CoV-2 infection. In addition, mice immunized with S-V987H show accelerated viral clearance in tissues compared with mice immunized with RBD or S-2P. The authors suggest that the S-V987H protein represents a promising alternative to S-2P for future SARS-CoV-2 vaccine development. Overall, the work takes a straightforward approach with a clear objective, and the results are concisely presented following standard experimental practices. However, I have several concerns that should be addressed in revising the manuscript.

COMMENTS

1. Although the spike protein used in this study was intentionally derived from the Wuhan-like strain of SARS-CoV-2, it would be valuable to demonstrate the applicability of the V987H mutation to Omicron variants, particularly the BA.5 spike included in the bivalent vaccine. Currently, spike proteins derived from the Omicron variant are known to have a lower yield compared to the original strain. Therefore, it would increase the importance of this manuscript if the potential of the V987H mutation to increase the yield of the Omicron spike could be explored and presented.

Following the reviewer's suggestion, we have evaluated the impact of incorporating the V987H mutation into the BA.5 background. In agreement with the reviewer's comment, we observed that the S-2P Omicron BA.5 protein was produced with lower yield than the S-2P WH1 protein. The incorporation of the V987H mutation did not improve the yield of the Omicron BA.5 Spike glycoprotein. These results are not surprising since V987H was selected based on WH1 background and Omicron BA.5 accumulate a larger set of mutations. Although the V987H cannot directly be implemented into the BA.5 background, further analysis interrogating additional positions surrounding the 987 amino acid might identify alternative mutations that might be functional equivalents of the V987H one.

Expression of S-2P (WH1), S-2P (BA5) and V987H (BA5) in Expi293 cells by transient transfection. Proteins were quantified by ELISA as it is indicated in the manuscript. Results are shown in µg/mL.

2. The S-2P variant was improved by the Hexa-Pro approach (DOI: 10.1126/science.abd0826), which involved the introduction of six prolines instead of two. It is conceivable that Hexa-Pro could replace S-2P in future spike protein designs. The authors mentioned briefly in the discussion but it is still not clear. To further strengthen the manuscript, the authors might consider addressing this point by comparing the results of their study with the yields reported in the literature based on the Hexa-Pro strategy.

The Hexa-Pro S protein incorporates four additional proline mutations, improving protein stability and yield by 10X. Hexa-Pro S protein was successfully incorporated in a Newcastle disease virus (NDV), which resulted in highly immunogenic and protected animal models from SARS-CoV-2 induced disease. However, no comparison with the S-2P protein was performed. Two studies have addressed the comparison between S-2P and S-6P in terms of immunogenicity and efficacy protecting animals from disease development. While Kalnin et al (10.1038/s41541-021-00324-5) observed that the immunogenicity of the S-2P immunogen was superior to the hexa-pro when they were assayed as mRNA vaccine, Lu and collaborators (10.1073/pnas.2110105119) described a higher immunogenicity and efficacy of the hexa-Pro protein compared with the S-2P when they were expressed in a VSV replicative vector. Interestingly, in the latest study, the authors showed that S-6P incorporate five times more efficiently on VSV particles than the S-2P protein. Thus, it is possible that the observed improvement was due to a higher amount of S-6P protein and not because this protein showed improved antigenicity. Here, we have used the same amount of S-2P and V987H recombinant proteins. Therefore, the differences observed in our study would mainly be due to differences in S antigenicity. We have addressed this point in the discussion section (lines: 525-533).

3. Please provide more details in the manuscript about the experiment shown in Fig. 1D. The Material and Methods section does not provide sufficient information about this experiment. In line 133, the authors mention that "most" of the variants are in a closed trimer conformation. Please indicate which variants in the graph are considered to be in this particular conformation.

Further information about how RBD exposure was determined has been added in the material and methods section (Lines 591-601). We considered all variants showing lower values of RBD exposure index than S-2P to be closer than the comparator.

Particularly, those with very low values: S-D985L, S-G416R, S-2M and S-5M

4. It would be useful to provide an explanation, or at least a comment, as to why replacing histidine with proline at amino acid position 987 might improve protein expression. In addition, it would be interesting to investigate whether the presence of K986P could affect the protein expression and/or immunogenicity of V987H.

Following the reviewer's suggestion, we have produced a novel S variant combining the K986P and the V987H mutations. The results showed that the double mutated variant was produced ten-fold lower than the S-2P protein. For additional information regarding to the role of V987H mutation, please see our reply to reviewer 1.

5. Please elaborate in their manuscript on the rationale for using DNA prime as opposed to administering two doses of recombinant proteins. It is imperative to clarify whether the inclusion of DNA prime could introduce confounding factors when comparing the immunogenicity of the two spike constructs.

The immunization approach has a major impact on the elicited immune response. Thus, the route of immunization, the adjuvant system and how antigens are presented to the immune system must be carefully selected. We think that both neutralizing antibodies and T cell responses are needed to confer protection. Thus, we chose immunization strategies that are known to elicit both immune responses. While DNA immunization generates T and B cell responses (doi:10.1038/nrg2432), heterologous DNA-prime/protein-boost strategies have shown superior results in many cases when compared with homologous regimens (10.1016/j.coi.2009.05.016). In a second approach, we decided to test a MF59-like adjuvant, since it is well known that this adjuvant induces both T and humoral responses (10.7554/eLife.52687), and it is used in vaccine for elderly, such as flu vaccine (10.1093/cid/ciac302). Our results with Addavax confirmed the previous results obtained with the DNA/protein-Aluminum immunization. Our aim was not to perform a head-to-head comparison between different immunization approaches, but to investigate whether the immunogenicity and efficacy outcome depended or not on a specific immunization regimen or adjuvant. Since both S-2P and S-V987H showed equivalent immunogenicity using different immunization approaches, we do not think DNA vaccination may be considered as major confounding factor in our study. We think that testing several adjuvant systems and different animal species to cross-validate immunogenicity and efficacy results add translational value. We have included some information to explain

our rationale for using the DNA-prime/protein-boost in lines 148-149 as well as AddaVax in lines 333-334.

6. Please consider including measurements of infection titers such as TCID50 or PFU in addition to the current methodology, which relies solely on viral genome copies to determine viral load. This data would allow for a more comprehensive assessment of viral replication dynamics.

Following the reviewer's suggestion we have added infection titers (TCID50). Please see Figure 3b, Figure 5b, Figure 7b and Supplementary Figure 5d.

MINOR COMMENTS

1. Please edit the title to avoid confusion that K18-hACE2 are mice, not hamsters.

The title has been edited as follows:

Immunization with V987H-stabilized Spike glycoprotein protects K18-hACE2 mice and golden Syrian hamster upon SARS-CoV-2 infection

2. Coomassie blue staining of the recombinant S constructs in Fig. 2A indicates higher expression of S-2P than of S-V987H. Please clarify whether the same amount of protein was loaded.

No, we do not load the same quantity of protein. Coomassie stained is just to show integrity and purity of the immunogens used during immunization. Quantification of protein was performed by ELISA, as indicated in the method section.

3. Please confirm that the unit of anti-RBD IgG (in AU/ml) in Fig.2C ranges from 0.1 to 10. This seems to be very low. Is it possible to convert it to standard units to compare with other studies?

ELISA units are in arbitrary units per mL. We cannot convert them to standard units because we do not have a proper standard to do this conversion.

REVIEWERS' COMMENTS

Reviewer #1 (Remarks to the Author):

In this revised manuscript, the authors presented data of additional experiments and tried to address the comments of the reviewers. The revised manuscript and the response have addressed most of the previous concerns, except on my previous comment point 8 about the conclusion that S-V987H conferring better protection than S-2P group against Beta VOC. Although supplementary experiments generate data about the T cell response of the vaccinated mice, it didn't lead to the conclusion stated in line 402-405, "These antibodies, along with other immune mediated effector functions, such as antibody dependent cellular phagocytosis or cytotoxicity, and T cell responses, might contribute to the increase in protection observed in S-V987H vaccinated animals, compared to the S-2P group." Of note, the difference between T cells responses of S-V987H and S-2P vaccinated mice was insignificant. I would be more convinced if it is concluded that S-V987H induced comparable immune protection as S-2P and was significantly better than the RBD control. Or it would be safer to state that the S-V987H vaccinated mice was better protected than the S-2P vaccinated mice, but the underlying mechanism of protection requires further investigation.

Reviewer #2 (Remarks to the Author):

Authors have satisfactorily addressed my major comments.

A section in the discussion on limitations of the presented data would be a good addition to the manuscript.

Reviewer #3 (Remarks to the Author):

The authors have addressed my comments well. However, I would like to see the discussion regarding the enhancement of protein expression that was observed in the Wuhan-like spike but not in Omicron's (BA.5), as the authors responded in the first comment. This is considered a limitation of this study and it must be noted.

REVIEWERS' COMMENTS

Reviewer #1 (Remarks to the Author):

In this revised manuscript, the authors presented data of additional experiments and tried to address the comments of the reviewers. The revised manuscript and the response have addressed most of the previous concerns, except on my previous comment point 8 about the conclusion that S-V987H conferring better protection than S-2P group against Beta VOC. Although supplementary experiments generate data about the T cell response of the vaccinated mice, it didn't lead to the conclusion stated in line 402-405, "These antibodies, along with other immune mediated effector functions, such as antibody dependent cellular phagocytosis or cytotoxicity, and T cell responses, might contribute to the increase in protection observed in S-V987H vaccinated animals, compared to the S-2P group." Of note, the difference between T cells responses of S-V987H and S-2P vaccinated mice was insignificant.

I would be more convinced if it is concluded that S-V987H induced comparable immune protection as S-2P and was significantly better than the RBD control. Or it would be safer to state that the S-V987H vaccinated mice was better protected than the S-2P vaccinated mice, but the underlying mechanism of protection requires further investigation.

We would like to thank to the reviewer for revising our manuscript. We have incorporated the following sentence in the newest version of the manuscript (lines 402-404).

"However, the underlying mechanisms that conferred the slight increase in protection observed in S-V987H vaccinated animals compared to the S-2P group needs further investigation."

Reviewer #2 (Remarks to the Author):

Authors have satisfactorily addressed my major comments.

A section in the discussion on limitations of the presented data would be a good addition to the manuscript.

We would like to thank to the reviewer for revising our manuscript. Following the referee suggestion, a limitations section has been added at the end of the discussion (lines 537-543).

Reviewer #3 (Remarks to the Author):

The authors have addressed my comments well. However, I would like to see the discussion regarding the enhancement of protein expression that was observed in the Wuhan-like spike but not in Omicron's (BA.5), as the authors responded in the first comment. This is considered a limitation of this study and it must be noted.

We would like to thank to the reviewer for revising our manuscript. This information has been incorporated in the limitations section added at the end of the discussion (lines 537-543).